



# Technical Note: Reanalysis of Aura MLS Chemical Observations

Quentin Errera[1], Simon Chabrillat[1], Yves Christophe[1], Jonas Debosscher[1], Daan Hubert[1], William Lahoz[2,†], Michelle L. Santee[3], Masato Shiotani[4], Sergey Skachko[5], Thomas von Clarmann[6], and Kaley Walker[7]

[1]Royal Belgian Institute for Space Aeronomy (BIRA-IASB), Brussels, Belgium
[2]Norsk Institutt for Luftforskning, NILU, Norway
[3]Jet Propulsion Laboratory, California Institute of Technology, USA
[4]Research Institute for Sustainable Humanosphere, Kyoto University, Japan
[5]Environment and Climate Change Canada (ECCC), Dorval, Quebec, Canada
[6]Karlsruhe Institute of Technology, Institute of Meteorology and Climate Research, Germany
[7]Department of Physics, University of Toronto, Canada
[†]Deceased, 1 April 2019

**Correspondence:** Quentin Errera (quentin.errera@aeronomie.be)

**Abstract.** This paper presents a reanalysis of the atmospheric chemical composition from the upper troposphere to the lower mesosphere from August 2004 to December 2017. This reanalysis is produced by the Belgian Assimilation System for Chemical ObsErvations (BASCOE) constrained by the chemical observations from the Microwave Limb Sounder (MLS) onboard the Aura satellite. BASCOE is based on the Ensemble Kalman Filter (EnKF) method and includes a chemical transport model driven by the winds and temperature from the ERA-Interim meteorological reanalysis. The model resolution is 3.75° in longitude, 2.5° in latitude and 37 vertical levels from the surface to 0.1 hPa with 25 levels above 100 hPa. The outputs are provided every 6 hours. This reanalysis is called BRAM2 for BASCOE Reanalysis of Aura MLS, version 2.

Vertical profiles of eight species from MLS version 4 are assimilated and are evaluated in this paper: ozone ($O_3$), water vapour ($H_2O$), nitrous oxide ($N_2O$), nitric acid ($HNO_3$), hydrogen chloride (HCl), chlorine oxide (ClO), methyl chloride ($CH_3Cl$) and carbon monoxide (CO). They are evaluated using independent observations from the Atmospheric Chemistry Experiment Fourier Transform Spectrometer (ACE-FTS), the Michelson Interferometer for Passive Atmospheric Sounding (MIPAS), the Superconducting Submillimeter-Wave Limb-Emission Sounder (SMILES), $N_2O$ observations from another MLS radiometer than the one used to deliver the standard product and ozonesondes. The evaluation is done in four regions of interest where only selected species are evaluated. These regions are (1) the lower stratospheric polar vortex where $O_3$, $H_2O$, $N_2O$, $HNO_3$, HCl and ClO are evaluated, (2) the upper stratospheric lower mesospheric polar vortex where $H_2O$, $N_2O$, $HNO_3$ and



CO are evaluated, (3) the tropical tropopause layer (TTL) where $O_3$, $H_2O$, CO and $CH_3Cl$ are evaluated and (4) the middle stratosphere where $O_3$, $H_2O$, $N_2O$, $HNO_3$, HCl, ClO and $CH_3Cl$ are evaluated.

In general BRAM2 reproduces MLS observations within their uncertainties and agrees well with independent observations, with several limitations discussed in this paper (see the summary in Sect. 5.5). In particular, ozone is not assimilated at altitudes above (i.e. pressures lower than) 4 hPa due to a model bias that cannot be corrected by the assimilation. MLS ozone profiles display unphysical oscillations in the TTL which are corrected by the assimilation, allowing a good agreement with ozonesondes. Moreover, in the upper troposphere, comparison of BRAM2 with MLS and independent observations suggests a positive bias in MLS $O_3$ and a negative bias in MLS $H_2O$. The reanalysis also reveals a drift in MLS $N_2O$ against independent observations which highlights the potential use of BRAM2 to estimate biases between instruments. BRAM2 is publicly available and will be extended to assimilate MLS observations post 2017.

©c Author(s) 2019. CC BY 4.0 License.



## 1 Introduction

Atmospheric reanalysis is an estimation of the past atmospheric state using the information provided by an atmospheric numerical model and a set of observations combined by a data assimilation system. Historically, atmospheric reanalyses have been produced by meteorological centres and upper-level products consisted mainly of temperature, winds, humidity, geopotential height and ozone. They have been used, e.g., "to understand atmospheric processes and variability, to validate chemistry-climate models and to evaluate the climate change" (Fujiwara et al., 2017).

With the increase of the number of chemical observations from satellites and the advent of chemical data assimilation systems (Lahoz and Errera, 2010), several reanalyses of the atmospheric chemical composition have been produced, most recently, the Copernicus Atmospheric Monitoring Service (CAMS) reanalysis (Inness et al., 2019) and the Tropospheric Chemical Reanalysis (TCR, Miyazaki et al., 2015). These two reanalyses focus mainly on the tropospheric composition with few assimilated species in the stratosphere (ozone in both cases and and nitric acid in TCR). With a focus on the stratosphere, Errera et al. (2008)

presented an assimilation of measurements from the Michelson Interferometer for Passive Atmospheric Sounding (MIPAS) but limited to only 18 months and to two species (ozone and nitrogen dioxide). Also focusing on the stratosphere, Viscardy et al. (2010) made an assimilation of observations from the Microwave Limb Sounder (MLS) onboard the UARS satellite between 1992 and 1997, but also limited to ozone.

The second generation of MLS, onboard the Aura satellite, is operating since August 2004 and is still measuring at the time of writing. It measures vertical profiles of around fifteen chemical species from the upper troposphere to the mesosphere with a high stability in terms of spatial and temporal coverage (SPARC/IO3C/GAW, 2019), and data quality (Hubert et al., 2016) . A subset of the Aura MLS (hereafter simply denoted MLS) constituents are assimilated in Near Real Time (NRT) since 2009 by the Belgian Assimilation System for Chemical ObsErvations (BASCOE) in order to evaluate the stratospheric

products from CAMS (Lefever et al., 2015). The BASCOE MLS analyses have also been used by the World Meteorological Organisation (WMO) Global Atmospheric Watch (GAW) program to evaluate the state of the stratosphere during polar winters (e.g. Braathen, 2016).



Chemical analyses of the stratosphere have additional potential applications. They could be used to evaluate chemistry-climate models. Usually, this is done with climatologies which are based on zonal mean monthly means of observations (Froidevaux et al., 2015; Davis et al., 2016; SPARC/IO3C/GAW, 2019) and thus affected by uncertainties due to the irregular sampling of the instruments, especially those with a low spatial coverage such as solar occulation instruments (Toohey and von Clarmann, 2013; Millán et al., 2016). Chemical analyses could also be used to study the differences between instruments using

the reanalysis as a transfer function (Errera et al., 2008). Moreover, chemical analyses could provide an internally consistent set of species to enable scientific questions to be addressed more completely than with measurements alone. Although not addressed in this paper, the data assimilation system provides the complete set of chlorine species while only a few of them are assimilated (hydrogen chloride and chlorine oxide) which can be useful to analyse polar processing studies. Finally, chemical analyses can be used to set model boundary conditions, e.g. the lower stratosphere in the estimation of carbon monoxide

emissions with inversion method (Müller et al., 2018).

On the other hand, the BASCOE-NRT analyses have several shortcomings, in particular the versions of the BASCOE system and of the MLS observations have changed several times since the start of the service. This paper thus presents a reanalysis of Aura MLS using one of the latest versions of BASCOE and MLS and covers the period August 2004-December 2017. Eight MLS species are assimilated: ozone ($O_3$), water vapour ($H_2O$), nitrous oxide ($N_2O$), nitric acid ($HNO_3$), hydrogen chloride

(HCl), chlorine oxide (ClO), methyl chloride ($CH_3Cl$) and carbon monoxide (CO). Although several other satellite instruments also measured vertical profiles of chemical stratospheric species during that period and beyond (see, e.g., SPARC/IO3C/GAW, 2019), these observations were not assimilated in order to avoid the introduction of spurious discontinuities such as in ERA-interim upper stratospheric temperature (Simmons et al., 2014, their Fig. 21).

The reanalysis presented in this paper is named the BASCOE Reanalysis of Aura MLS, version 2 (BRAM2). The version 1

of the reanalysis, BRAM1, has been released but not published. This paper is organized as follows. Section 2 presents the MLS observations assimilated in BRAM2 and the independent observations used for its validation. Section 3 presents the BASCOE system and its configuration for BRAM2. The method to intercompare BRAM2 with the observations is described in Sect. 4. The evaluation of BRAM2 is presented in Sect. 5, including a summary. The conclusions are given in Sect. 6.



## 2 Observations

### 2.1 The assimilated MLS observations

The BRAM2 reanalysis is based on the assimilation of observations taken by the Microwave Limb Sounder (MLS) operating on NASA's Aura satellite. MLS measures vertical profiles of around fifteen chemical species. For BRAM2, the following species have been assimilated: $O_3$, $H_2O$, $N_2O$, $HNO_3$, $HCl$, $ClO$, $CO$ and $CH_3Cl$. Other MLS species are not considered because,

with the exception of OH, they either are available over only a limited vertical range or require substantial averaging prior to use in scientific studies. OH profiles have not been assimilated because modeled OH is more controlled by the atmospheric conditions (e.g. temperature) and the state of long-lived species (in particular $H_2O$) than by its initial conditions. While a similar situation holds for ClO in the middle stratosphere (here the long-lived species would be HCl), this is not the case in conditions of chlorine activation, such as in the lower stratospheric polar vortex.

MLS was launched in July 2004 and provided its first profiles in August of that year. At the time of writing, the instrument is still in operation despite showing some aging degradation. Around 3500 vertical profiles are delivered every day, measured during day and night time. In this paper, we have used version 4.2 (v4) of MLS profiles as described in Livesey et al. (2015, denoted L2015 hereafter). Each MLS profile is checked before assimilation according to the recommendations given in L2015. Profiles are only assimilated in the vertical range of validity given in L2015 and reported in Table 1. Profiles, or part of them,

are discarded if the "Estimated Precision", "Quality", "Convergence" and "Status" are outside the ranges given in L2015. In particular, this screening discarded profiles contaminated by clouds, mainly for $O_3$, $HNO_3$, and CO. ClO profiles show biases at and below 68 hPa and have been corrected according to L2015.

MLS $O_3$ profiles exhibit vertical oscillations in the Tropical Tropopause Layer (TTL, see L2015, Yan et al., 2016). Although improvements have been made in v4 compared to previous versions, this problem has not been eliminated (see Sect. 5.4). The

BASCOE CTM also suffers from an ozone deficit around 1 hPa that assimilation cannot correct. This led us to assimilate MLS $O_3$ observations only at altitudes below (i.e. pressure greater than) 4 hPa (see Sect. 3.1).

In MLS v4, the standard product for $N_2O$ is derived from radiances measured by the 190-GHz radiometer. Previous MLS data versions used the 640-GHz radiometer, which provided slightly better quality, but this product ceased to be delivered after



August 2013 because of instrumental degradation in the band used for that retrieval. For BRAM2, the 190-GHz $N_2O$ product

is assimilated for the whole period to avoid discontinuity when switching between different products.

CO profiles suffer from several artifacts as reported by L2015. They show a positive systematic error of 20-50% in the

mesosphere and a negative systematic error of 50-70% near 30 hPa. Between 1 and 0.1 hPa, profiles are rather jagged. There

is also a tendency for negative values below levels where CO abundances are large, especially in the polar vortex when high

concentrations of CO descend to the mid-stratosphere. No corrections have been applied to resolve these artifacts because none

are recommended by the MLS team. Although BRAM2 has assimilated MLS CO within its recommended vertical range of

validity (0.0046-215 hPa), BRAM2 CO will be evaluated only where CO is relevant for stratospheric dynamics, i.e., in the

polar vortex above 10 hPa and in the TTL.

The error budget of each species has also been estimated by L2015. This information is given as uncertainty profiles of

accuracy and precision, and will be used in the validation of the BRAM2 products. Note that L2015 provides the 2-$\sigma$ accuracy

and 1-$\sigma$ precision. In this paper, we are using the 1-$\sigma$ uncertainties for both the accuracy and precision.

## 2.2   Independent observations used for validation

### 2.2.1   ACE-FTS

The Atmospheric Chemistry Experiment Fourier Transform Spectrometer (ACE-FTS, Bernath et al., 2005) performs infrared

solar occultation measurements of the atmosphere. It has been in operation since February 2004 and continues to make routine

measurements. Its inclined circular orbit provides up to 30 measurements (sunrise and sunset) per day with a focus on the

high latitudes. We used the ACE-FTS version 3.6 dataset which provides profiles of temperature and more than 30 trace gases

(Boone et al., 2013). The vertical resolution of these measurements is  3 km based on the instrument field-of-view (Boone

et al., 2005).

The ACE-FTS v3.6 profiles of $O_3$, $H_2O$, $N_2O$, $HNO_3$ and CO have recently been validated through comparisons with MLS

and MIPAS (Sheese et al., 2017). For HCl, validation studies for the previous ACE-FTS version (v2.2) were done by Mahieu

et al. (2008) and Froidevaux et al. (2008). Additional comparisons of HCl using ACE-FTS v3 have been made with SMILES

measurements (Sugita et al., 2013). The differences between the ACE-FTS v2.2 and v3 datasets were presented by (Waymark et al., 2013). Measurements of $CH_3Cl$ from ACE-FTS and MLS have been compared by Santee et al. (2013). These results are used to provide profile uncertainties for this study. Currently, ClO is a research product for ACE-FTS and is not part of the standard v3.6 data set. Thus, it is not used in the comparisons with BRAM2. All ACE-FTS data used in this study were screened using the version 2.1 quality flags algorithm.

### 2.2.2 MIPAS

The Michelson Interferometer for Passive Atmospheric Sounding (MIPAS, Fischer et al., 2008) was a limb-viewing spectrometer recording mid-infrared spectral radiances emitted by the atmosphere. MIPAS was part of the Envisat instrumentation, operating between July 2002 and April 2012. Its sun-synchronous low-Earth orbit allowed relatively dense global coverage during day and night time with about 1080 to 1400 profile measurements per day, depending on the observation mode. The MIPAS mission is divided in two phases: the full-resolution phase from 2002 to 2004 and the optimised-resolution phase from 2005 to 2012. The latter period is characterised by finer vertical and horizontal sampling attained through a reduction of the spectral resolution. In this study, MIPAS data from the second phase have been used, i.e. from 2005 to 2012.

MIPAS spectral radiance measurements were used to derive vertical profiles of temperature and trace gas concentrations. In this study, we used trace gas concentrations produced with the data processor developed and operated by the Institute of Meteorology and Climate Research (IMK) in cooperation with the Instituto de Astrofísica de Andalucía (IAA-CSIC) (von Clarmann et al., 2003). Updates of the data processing scheme, relevant for more recent data versions, are reported in von Clarmann et al. (2009, 2013). The latter paper documents the data versions used here, namely V5_[product_name]_22[0_or_1][1]. The MIPAS ozone product was thoroughly investigated within the European Space Agency's Climate Change Initiative (Laeng et al., 2014). The MIPAS water vapour product has been validated within the framework of the Stratosphere-troposphere Processes And their Role in Climate (SPARC) Water Vapor Assessment activity (Lossow et al., 2017, 2018). A high bias in the lower part of MIPAS $N_2O$ retrievals is discussed and partly remedied by Plieninger et al. (2015). The retrieval scheme for CO was developed by Funke et al. (2009). MIPAS $O_3$, $H_2O$, $N_2O$, $HNO_3$ and CO profiles were compared to those of ACE-FTS

---

[1]Versions 220 and 221 are equivalent from the data user perspective; these different version numbers shall ensure traceability with respect to technical details.

and MLS by Sheese et al. (2017). The ClO retrieval was originally developed for MIPAS full spectral resolution measurements of the years 2002-2004 (Glatthor et al., 2004). The application to the reduced spectral resolution phase of the years 2005-2012, used in this work, led to unrealistic values in the upper stratosphere, a problem that has been fixed only for more recent data versions. Thus, MIPAS ClO will only be used during conditions of chlorine activation in the polar winters.

### 2.2.3 Ozonesondes

In-situ measurements of ozone between the surface and 30-35 km altitude are performed routinely by small meteorological balloons launched two to four times a month at several tens of stations around the globe. Such balloons are equipped with a radiosonde that records ambient pressure, temperature and relative humidity, a GPS sensor which geolocates each measurement in 3+1 dimensions, and an ozonesonde which registers ozone partial pressure. The typical vertical resolution of the measurements is 100-150 m (Smit and the Panel for the Assessment of Standard Operating Procedures for Ozonesondes, 2014; Deshler et al., 2017). Uncertainties are assumed random and uncorrelated (Sterling et al., 2018) and are around 5% in the stratosphere, 7-25% around the tropopause and 5-10% in the troposphere. This study considers the sonde data collected at 33 stations of the Network for the Detection of Atmospheric Composition Change (NDACC).

Ozone profiles (and the associated temperature) have been smoothed in order to limit the number of points per profile, which is often larger than 1000. This is done by averaging the measurements on a 100 m vertical grid.

### 2.2.4 SMILES ClO

The Superconducting Submillimeter-Wave Limb-Emission Sounder (SMILES) onboard the International Space Station (ISS) monitored the global distribution of minor constituents of the middle atmosphere from October 2009 to April 2010. It was developed to demonstrate the high sensitivity of the 4-K cooled submillimeter limb sounder in the environment of outer space (Kikuchi et al., 2010). The total number of profiles per day was about 1600. We used the SMILES Level 2 (L2) data v2.4 processed by the Japan Aerospace Exploration Agency (JAXA, Mitsuda et al., 2011; Takahashi et al., 2010, 2011), providing vertical profiles of minor atmospheric constituents (e.g. $O_3$ with isotopes, HCl, ClO, $HO_2$, BrO, and $HNO_3$). SMILES



L2 JAXA products and some related documents including a Product Guide for each version were released for public use (https://doi.org/10.17597/ISAS.DARTS/STP-00001).

There are several research papers dealing with SMILES chlorine related species. Akiyoshi et al. (2016) investigated the chemical constituent distributions during the major stratospheric sudden warming in the northern winter of 2009/2010 by the use of a chemistry climate model simulation nudged towards a meteorological reanalysis. The results were compared with

SMILES and MLS observations. They found that the evolution and distribution of ozone and HCl inside/outside the polar vortex associated with the vortex shift to midlatitudes in January are quite similar between the two instruments. Those of ClO are also similar, considering the difference in the local time of the measurement. Sugita et al. (2013) also compared SMILES ClO profiles inside the Antarctic vortex in November 2009 with MLS and found an agreement around $\pm 0.05$ ppbv for ClO abundance less than 0.2 ppbv.

**3   The BASCOE system and its configuration for BRAM2**

**3.1   BASCOE**

The BRAM2 reanalysis has been produced by the assimilation of MLS observations using the Belgian Assimilation System for Chemical ObsErvations (BASCOE, Errera et al., 2008; Errera and Ménard, 2012; Skachko et al., 2014, 2016). The system is based on a chemistry transport model (CTM) dedicated to stratospheric composition which includes 58 chemical species.

For BRAM2, dynamical fields are taken from the European Centre for Medium-Range Weather Forecasts (ECMWF) ERA-Interim reanalysis (Dee et al., 2011). The model horizontal resolution is $3.75°$ longitude $\times$ $2.5°$ latitude. The vertical grid is represented by 37 hybrid pressure levels going from the surface to 0.1 hPa which are a subset of the ERA-Interim 60 levels. ERA-Interim is preprocessed to the BASCOE resolution ensuring mass flux conservation (Chabrillat et al., 2018). The model time step is 30 minutes. All species are advected by the flux-form semi-Lagrangian scheme (Lin and Rood, 1996). Around 200

chemical reactions (gas-phase, photolysis and heterogeneous) are taken into account and the gas-phase and photolysis reaction rates have been updated according to Burkholder et al. (2011).

As many other models (SPARC, 2010, see their Fig. 6.17), the BASCOE model suffers from an ozone deficit in the upper stratosphere lower mesosphere. Skachko et al. (2016) showed that around 1 hPa, BASCOE underestimates MLS ozone by ~20%. They also pointed out that this deficit cannot be corrected by the assimilation of observations because the ozone lifetime is much shorter than the revisit time of MLS, typically 12 hours between an ascending and a descending orbit of the Aura satellite. It turns out that assimilation of ozone in this altitude region introduces spatial discontinuities in the ozone fields

around the locations of the most recent observations. For this reason, MLS $O_3$ observations at altitude above 4 hPa have not been assimilated, and the BRAM2 ozone will not be discussed above that level.

The microphysics of Polar Stratospheric Clouds (PSCs) and their impact on the chemistry is taken into account by a simple parameterization as described in Huijnen et al. (2016) but with several updates. In its original implementation, the CTM overestimated the loss of HCl by heterogeneous chemistry (in contrast to other models which underestimate the loss of HCl,

see Wohltmann et al., 2017; Grooß et al., 2018). Preliminary experiments prior to BRAM2 showed that data assimilation was not able to correct for this bias (not shown). The parameters of Huijnen et al.'s formulation have been tuned by trial and error through CTM simulations of the Antarctic winter 2008. The best setup found includes the following updates: (1) nitric acid tri-hydrate (NAT) PSCs are assumed to exist when the ratio between $HNO_3$ vapor pressure and the equilibrium vapor pressure exceeds a supersaturation ratio set to 10 as in Considine et al. (2000), compared to 1 in the original setting; (2) the NAT surface

area density has been reduced from $2 \cdot 10^{-7}$ to $10^{-7}$ cm$^2$ cm$^{-3}$; (3) the characteristic timescale of NAT sedimentation has been reduced from 20 to 10 days. CTM results with this setup are discussed in Sect. 5.2.

Condensation of water vapour is approximated by capping its partial pressure to the vapour pressure of water ice (Murphy and Koop, 2005).

Two data assimilation methods have been implemented in BASCOE: 4D-Var (Errera and Ménard, 2012) and EnKF (Skachko

et al., 2014, 2016). BRAM2 uses the EnKF method because this implementation offers a better scalability than 4D-Var on cluster computers. EnKF provides an estimation of the analysis uncertainty based on the standard deviation of the ensemble state. These values have not been evaluated here and will be the subject of a future study. For this reason, the standard deviation of the ensemble is not provided in the BRAM2 dataset. The EnKF implementation in BASCOE cycles through the following steps:



1. At initial time, an ensemble of 20 members is generated based on 20% Gaussian perturbations of a given model initial state.

    2. Each ensemble member is propagated in time using the BASCOE CTM to the next model time step.

    3. If MLS observations are available at the current model time step, add a perturbation to each ensemble member (see Sect. 3.2).

4. Save the ensemble mean and its variance (see Sect. 3.4)

    5. If MLS observations are available, the EnKF equation is solved to compute the analysis for each ensemble member.

    6. Steps 2 to 5 are repeated until the last model time step is reached.

## 3.2  EnKF setup

BRAM2 is the result of four streams (or runs) that have been produced in parallel to reduce the production time. The first

stream starts on 1 August 2004, a few days before the first available MLS observations. The three next streams start on 1 April 2008, 2012 and 2016, respectively. Streams 1-3 end on 1 May 2008, 2012 and 2016, respectively, allowing one month of overlap between each stream. The fourth stream currently ends on 1 Jan 2018 and will be extended.

Initial conditions are taken from a 20-year BASCOE CTM simulation where boundary conditions for tropospheric source gases (e.g. $CH_4$, $N_2O$ or Chlorofluorocarbons - CFCs) vary as a function of latitude and time (Meinshausen et al., 2017). The

20 ensemble member states are calculated by adding spatially correlated perturbations to the initial conditions as described in Skachko et al. (2014).

The BASCOE setup used to produce BRAM2 is almost identical to the experiments performed by Skachko et al. (2016), since both studies assimilate the same observations with the same model. Horizontal and vertical localization length scales are defined as $L_h$=2000 km and $L_v$=1.5 model level, respectively. Note that correlations between the species are not taken into

account in BASCOE EnKF. Except for $HNO_3$, BASCOE uses a Background Quality Check (BgQC, Anderson and Järvinen, 1999; Skachko et al., 2016) which rejects any observation if its departure from the mean ensemble state is five times the

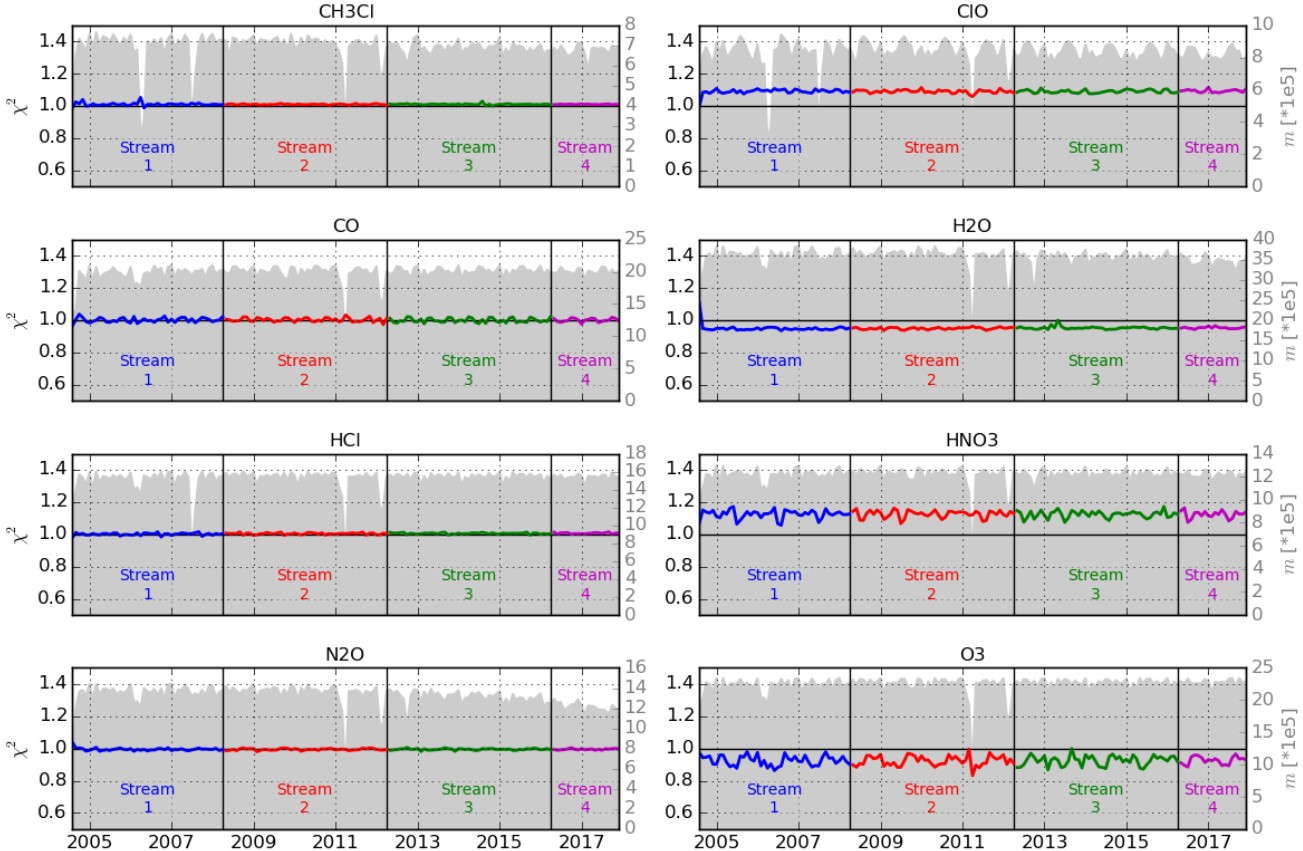

**Figure 1.** Time series of the monthly mean $\chi^2$ for MLS assimilated species (colored lines - one color for each stream, left y-axes) and the corresponding number of assimilated observations $m$ (gray bars, right y-axes). The horizontal black lines show the expected theoretical value of $\chi^2 = 1$ and the vertical black lines show the dates of the transition between the different streams.

combined error of the observations and the background. For $HNO_3$, the BgQC was turned off because preliminary experiments prior to BRAM2 have shown better Observation - minus - Forecast statistics without this setup.

The system includes two adjustable parameters that need to be calibrated: the model error parameter $\alpha$ and the observational error scaling factor $s_o$. The model error is calibrated using a $\chi^2$ test. At a given model time step $k$, $\chi_k^2$ measures the difference between the observations and the model forecast weighted by their combined error covariances. Ideally, if the covariances are correctly specified and if the model is un-biased, the average $\chi_k^2$ should be close to the number of observations $m_k$, i.e.

$\langle \chi_k^2 / m_k \rangle \sim 1$ where $\langle \rangle$ denotes the mathematical expectation. Ménard and Chang (2000) have shown that the slope in the time series of $\langle \chi_k^2 / m_k \rangle$ is sensitive to the model error parameter $\alpha$ while the time-average of $\langle \chi_k^2 / m_k \rangle$ is sensitive to the



observational error. For MLS assimilation using BASCOE, Skachko et al. (2014, 2016) found a single value of $\alpha = 2.5\%$ for each assimilated species and each model grid point, and the same value was used for BRAM2. For the observational error scaling factor, a vertical profile for each species has been calibrated using the Desroziers' method (Desroziers et al., 2005) as implemented in Skachko et al. (2016).

Figure 1 shows the time series of the monthly mean $\langle \chi_k^2/m_k \rangle$ for the four streams. The total number of monthly assimilated observations is also shown. For all species, the $\chi^2$ time series are stable, as expected. This validates the choice of $\alpha = 2.5\%$. For $CH_3Cl$, CO, HCl and $N_2O$, the values are very close to 1. For ClO and $HNO_3$, the values are slightly higher than 1 (around 1.1) and for $H_2O$ and $O_3$, the values are slightly lower than 1 (around 0.95 and 0.9, respectively). The $\chi^2$ time series for $HNO_3$ and $O_3$ also display seasonal variations of small amplitude (<0.1). Overall, these deviations are relatively small, e.g. when comparing with a $\chi^2$ test obtained by the Tropospheric Chemical Reanalysis (Miyazaki et al., 2015). This validates the implementation of Desroziers' method to adjust the observational error scaling factors. Note that the transition between the streams does not display visible discontinuities in the $\chi^2$ time series, which validates the choice of a one-month overlap between the streams. (Values for the first month of streams 2-4, which overlap with the last month of streams 1-3, are not shown.)

Figure 2 shows the time series of the observational error scaling factors estimated by Desroziers' method for all species at five selected pressure levels. Values higher (lower) than 1 indicates that the MLS error has been increased (decreased) by the Desroziers' method. Values are usually between 0.5 and 1.5 except for $O_3$ at 100 hPa, which has a value between 2 and 3, this being likely due to vertical oscillation in MLS $O_3$ profiles in the TTL (see Sect. 5.4). The time series display seasonal variations for some species and/or levels usually with a six month period attributed to both polar winter seasons. As for the $\chi^2$ test, no discontinuities are visible at the transition time between the streams. For some species and/or levels, the time series show a small positive drift. The cause of this drift has not been identified but is unlikely due to an issue in the BASCOE system. In such a case, this would have resulted in discontinuities at the dates of transition between the streams in the time series of the observational error scaling factors. This issue has not been investigated further in this paper.





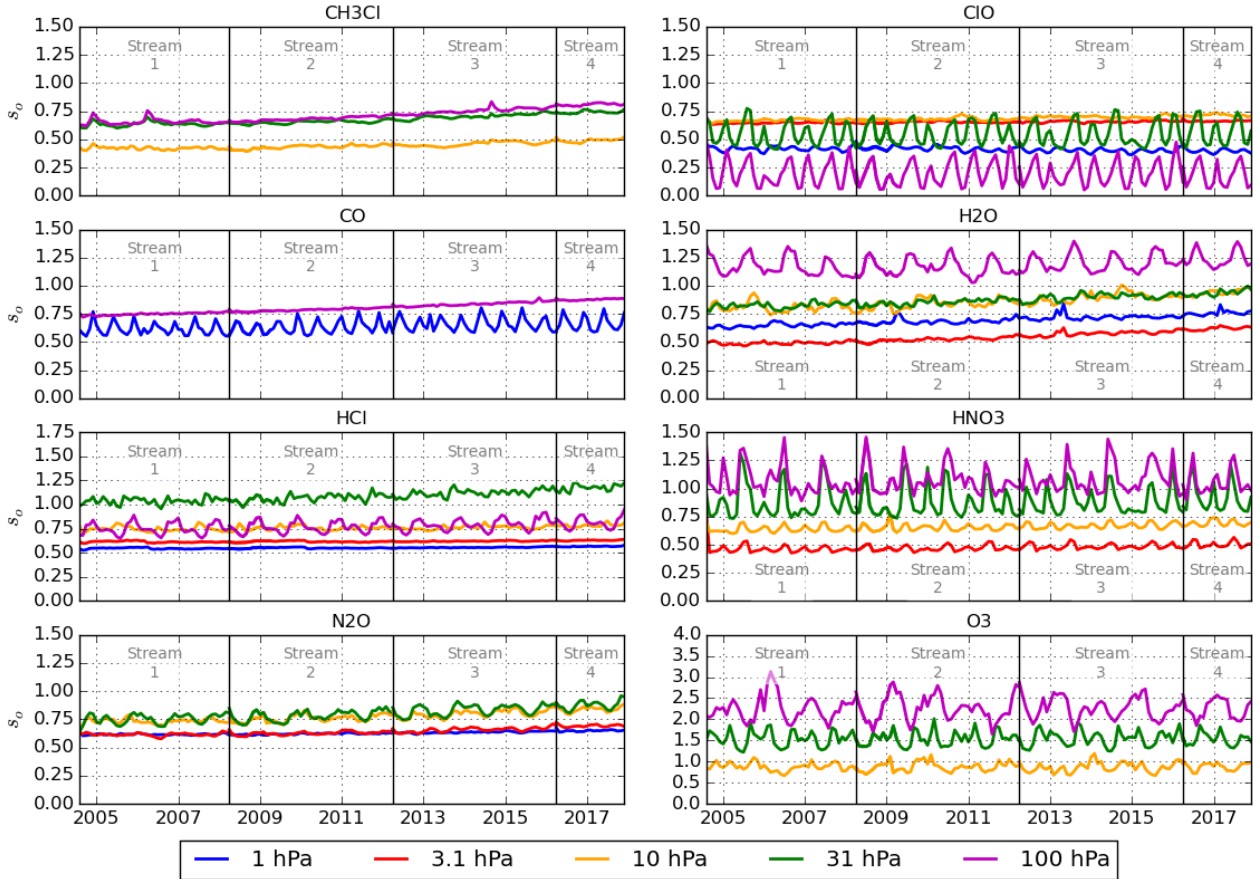

**Figure 2.** Time series of monthly mean observational error scaling factors $s_o$ for each assimilated species at five specific MLS levels: 1, 3.1, 10, 31 and 100 hPa. The vertical black lines show the dates of the transition between the different streams.

## 3.3 BASCOE Observation Operator

The observation operator of BASCOE consists of a linear interpolation of the model state to the geolocation of the observed profile points available at the model time $\pm$ 15 minutes, i.e. half of the model time step. It has been used to save the BRAM2 state in the space of MLS as well as in the space of the independent observations, except for NDACC ozonesondes, during the BRAM2 production. For NDACC ozonesondes, the BRAM2 state has been interpolated to the NDACC station from the

5   6-hourly BRAM2 gridded outputs. The error introduced by this method is negligible for $O_3$ below 10 hPa where ozonesondes are used (Geer et al., 2006). Note that no averaging kernels of any satellite dataset have been used in the BASCOE observation operator because the BASCOE EnKF is not ready for their use. The vertical resolution of these observations is sufficiently high



– and similar to the model vertical resolution – that their use is typically considered unnecessary. We will see, however, that this is not always the case (see Sect. 5).

### 3.4 BRAM2 Outputs

BRAM2 gridded outputs are the 6-hourly mean of the ensemble state. A second type of output is given in the space of the observations (see previous section) and will be referred to below as model-at-observation or, in short, ModAtObs. All outputs

(gridded and ModAtObs) are taken at step 4 of the assimilation cycle (see Sect. 3.1), i.e. corresponding to the background state. For the gridded outputs, this allows the last model forecast to smooth any discontinuities at the edge of regions influenced by observations. For ModAtObs outputs, this means that all comparisons between BRAM2 and observations shown in this paper are using the background state.

### 3.5 Control Run

For this publication, a control run has been produced, labeled CTRL. It is a BASCOE CTM simulation using the same configuration as BRAM2, covering the period May 2009 - Nov 2010, and initialized by the BRAM2 analysis. CTRL is used to evaluate the added value of the assimilation compared to a pure model run where an 18 months simulation is sufficiently long. It will also indicate model processes that need to be improved in the future.

### 4 Intercomparison method

The evaluation of BRAM2 is based on means and standard deviations of the differences between BRAM2 and the assimilated or the independent observations. In all cases, the BRAM2 forecast (or background) is used, i.e. at step 4 of the assimilation cycle (see Sect.3.1). These statistics are denoted forecast-minus-observations (FmO). FmO are calculated either in pressure/latitude or potential temperature/equivalent latitude domain. In the first case, the statistics are calculated on the MLS pressure grid or, for the other datasets which are given on a kilometric vertical grid, on pressure bins with 12 bins per decade of pressure

using the pressure profiles from these datasets. In the second case, all products are interpolated on potential temperature (theta) levels using their measured pressure and temperature profiles. Equivalent latitudes at the observations are interpolated from

ERA-Interim daily fields of potential vorticity, at 12 UT, calculated on a 1°x 1°latitude/longitude grid and with a 35 level theta

grid from 320 to 2800 K (Manney et al., 2007). Finally, statistics in % are normalized to the mean of the BRAM2 forecast

corresponding to the same period/region.

The FmO statistics will also be compared to the MLS error budget provided in the MLS data quality document (see L2015).

The mean and standard deviations of (BRAM2-MLS) differences will be compared, respectively, to the MLS accuracy and

precision. Depending on the species, these values are provided in volume mixing ratio (vmr), in %, or both. If necessary, the

conversion from % to vmr, and vice versa, will use MLS average observations corresponding to the shown situation.

In order to determine if the origin of the biases between BRAM2 and independent observations are due to MLS or the

BASCOE CTM, the FmO statistics have also been compared to the mean and standard deviation of the difference between

MLS and ACE-FTS as provided in other validation studies. These values have been digitized from Froidevaux et al. (2008) for

HCl, Santee et al. (2013) for $CH_3Cl$, Sheese et al. (2017, denote hereafter S2017) for $O_3$, $H_2O$, $N_2O$, $HNO_3$ and CO (for CO,

only during polar winter conditions). Error profiles from S2017 are converted from a kilometric to a pressure vertical grid using

a log-pressure altitude relationship with a scale height of 7 km. Santee et al. (2013) and S2017 also show the mean profiles

of MLS and ACFTS used to make their comparison, allowing us to convert from % to vmr. For Froidevaux et al. (2008), this

conversion is based on the average MLS observations corresponding to the shown situation.

**5   Evaluation of the Reanalysis**

Figure 3 displays the daily zonal means of MLS, BRAM2 and CTRL on 1 Sept 2009 for $O_3$, $H_2O$, $N_2O$, $HNO_3$, HCl, ClO

(daytime values), $CH_3Cl$ and CO. Only BRAM2 species constrained by MLS will be evaluated in this paper. This figure

highlights regions of good/poor qualitative agreement between BRAM2 and MLS and the added value of the assimilation

compared to a pure model run (CTRL). The figure also highlights regions with chemical or dynamical regimes that will be

explored in more detail in this section.

One of these regions is the lower stratosphere in the polar vortex (denoted hereafter LSPV) where PSC microphysics takes

place. In this region (between 10-100 hPa and 90°S-60°S in the figure), $HNO_3$ and $H_2O$ are lost due to PSC uptake and



**Figure 3.** Daily zonal means of MLS observations (left column) on 1 September 2009, the corresponding ModAtObs values of BRAM2 (center column) and CTRL (right column). From top to bottom: $O_3$ (ppmv), $H_2O$ (ppmv), $N_2O$ (ppbv), $HNO_3$ (ppbv), $HCl$ (ppbv), $ClO$ (ppbv, daytime values), $CH_3Cl$ (pptv) and CO (ppbv, note the log scale). Zonal means are calculated on the MLS pressure grid and binned on a 5° latitude grid. White squares in the MLS CO plot denote negative values. BRAM2 $O_3$ is not assimilated (and not shown) at altitude above 4 hPa, see text for details.

sedimentation, HCl is destroyed by heterogeneous chemistry on the surface of PSCs and ClO is produced. All these features, observed by MLS, are well reproduced by BRAM2. The comparison of BRAM2 and CTRL highlights some model deficiencies, e.g., the underestimation of $H_2O$ loss and ClO enhancement. The isolation of polar air from midlatitudes is also visible by the strong $N_2O$ horizontal gradient around 60°S, observed by MLS, well reproduced by BRAM2 and underestimated in CTRL.

Another region is the upper stratosphere lower mesosphere (USLM) in the polar vortex (hereafter denoted USPV). This

region (between 0.1-10 hPa and 90°S-60°S in the figure) is affected by the descent of mesospheric and thermospheric air rich in CO and poor in $H_2O$. BASCOE does not include upper boundary conditions for these sources and losses so CTRL displays much higher $H_2O$ and much lower CO than MLS in USPV. BRAM2 on the other hand agrees well with the observations.

The third identified region is the tropical Upper Troposphere Lower Stratosphere (UTLS), or Tropical Tropopause Layer (TTL between 70 to 300 hPa), where tropospheric source gases enter in the stratosphere. Relevant species in this region are

$H_2O$, $O_3$, $CH_3Cl$ and CO ($N_2O$ would have been relevant but the MLS $N_2O$ retrieval is not recommended for scientific use at pressures greater than – altitudes below – 68 hPa). BRAM2 agrees well with MLS in this region for these species. It improves the vertical gradient of $H_2O$ found in CTRL and the amount of CO and $CH_3Cl$. Note that BRAM2 will not be discussed in the extratropical UTLS.

The fourth region includes everything not in the LSPV, USPV and the UTLS regions. Since it covers most of the middle

stratosphere, it will be denoted MS. In this region, BRAM2 and MLS agree generally well, e.g., at the ozone peak, for the horizontal gradient of $N_2O$ and the vertical gradient of HCl.

Figure 4 shows the mean and standard deviation of the differences between BRAM2 and MLS, associated with their daily zonal mean shown in Fig. 3. These statistics are in % and are normalized by the mean of BRAM2, as will be the case for the rest of the paper. In general, the normalized mean and standard deviation of the differences are low where the abundance of the

species is relatively high, with the exception of $H_2O$ in the upper troposphere. Conversely, the (normalized) mean and standard deviation are high where the abundance of the species is low, i.e: (1) $O_3$ in the tropical troposphere, (2) $N_2O$ above 5 hPa and in the polar vortex, (3) $HNO_3$ in the TTL, above 5 hPa and in the polar vortex, (4) HCl in the TTL and in the polar vortex, (5) ClO in the lower stratosphere, (6) $CH_3Cl$ above 10 and 30 hPa in, respectively, the Tropics and the mid-latitudes and (7) CO in the MS and LSPV.



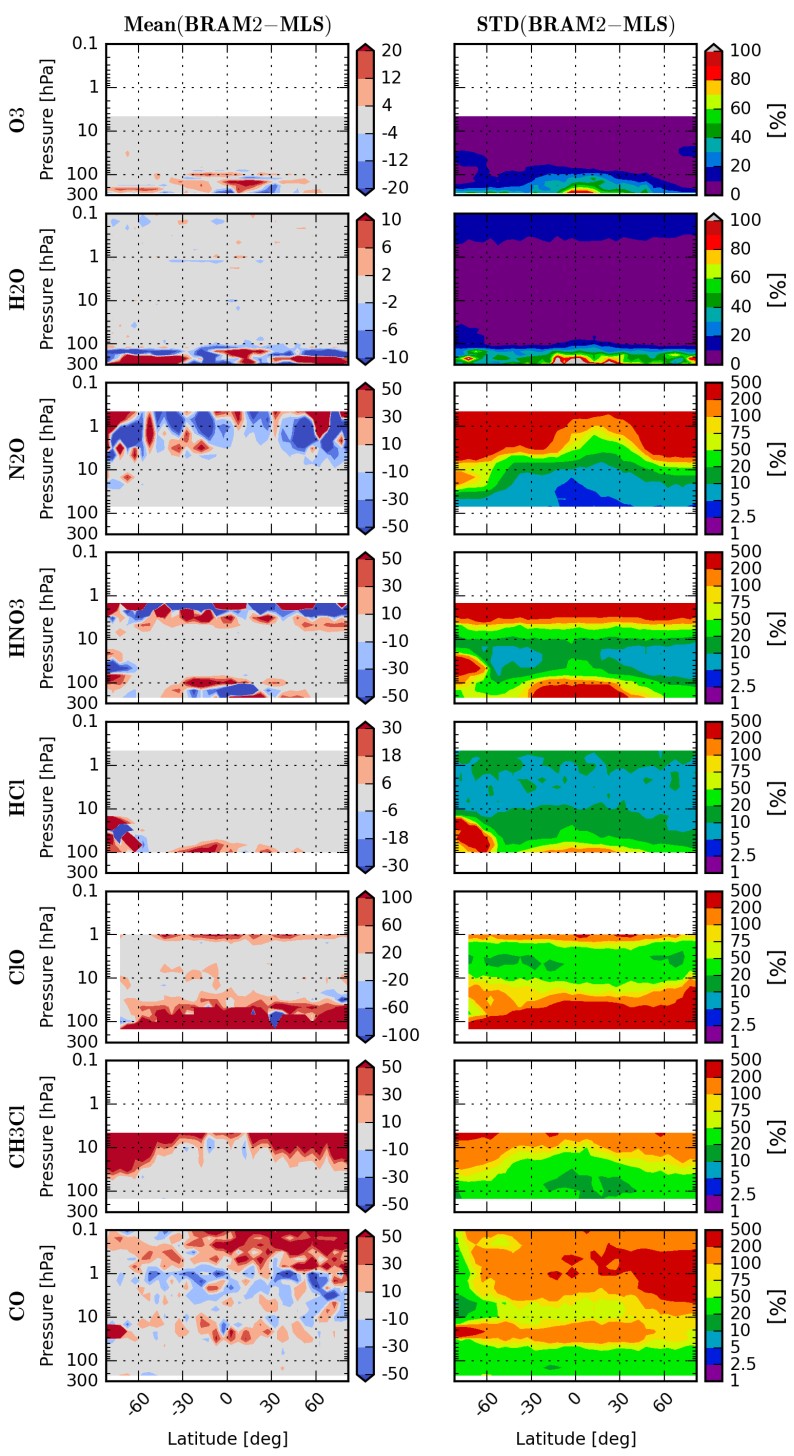

**Figure 4.** Daily zonal mean mean differences (in %) between BRAM2 and MLS (left column) and the associated standard deviation (right column) on 1 September 2009. From top to bottom: $O_3$, $H_2O$, $N_2O$, $HNO_3$, HCl, daytime ClO, $CH_3Cl$ and CO. Zonal means are calculated on the MLS pressure grid and binned on a 5° latitude grid. Note that the ranges in the colorbars differ for each individual plot and that some colorbars are in log scale.

In the following subsections, BRAM2 will be evaluated in the four above-mentioned regions: the Middle Stratosphere (MS), the Lower Stratospheric Polar Vortex (LSPV), the Upper Stratospheric lower mesospheric Polar Vortex (USPV) and the Tropical Tropopause Layer (TTL). The objective of these evaluations is to answer several questions. How well does BRAM2 agree with assimilated and independent observations? In which regions and altitudes is BRAM2 recommended for scientific use i.e. well characterized against independent observations with FmO statistics stable in time. For this evaluation, we have

used five well characterized sets of independent observations: ACE-FTS, MIPAS, SMILES ClO, MLS_N2O_640 (the other MLS $N_2O$ product retrieved from the 640 GHz radiometer which was turned off in July 2013) and ozonesondes. A summary of the evaluation is given in Sect. 5.5.

## 5.1    Middle Stratosphere (MS)

The evaluation of BRAM2 in the middle stratosphere is based on two figures: one showing vertical profiles of the FmO (Fig.5)

and the other showing time series of the FmO at selected pressure levels (Fig. 6). The first figure shows the mean and standard deviation of the FmO between BRAM2 and observations from MLS, ACE-FTS, MIPAS, MLS_N2O_640, SMILES ClO and ozonesondes. These statistics are calculated between 30°N-60°N and 0.1-100 hPa for the 2005-2017 period for MLS and ACE-FTS (FmO profiles between 60°S-30°S and 30°S-30°N are given in the supplement). For MIPAS and MLS_N2O_640, the datasets end in March 2012 and July 2013, respectively. For SMILES, the period is October 2009–April 2010. Note that

comparison with MIPAS ClO is only done in the polar winter conditions (see Sect. 2.2.2). CO is not shown in the figure because it is chemically irrelevant in the middle stratosphere – CO will be discussed in the USPV and the TTL subsections. The figure also shows two types of error: first, the MLS accuracy and precision which are compared, respectively, to the mean and the standard deviation of the differences; second, the mean and standard deviation of the differences between MLS and ACE-FTS (see Sect. 4).

The second figure (Fig. 6) shows time series of monthly FmO for the 2005-2017 period corresponding to 30°N-60°N latitude band at three pressure levels in the high, middle and lower stratosphere: 0.68, 4.6 and 46 hPa. Statistics shown are the bias against the different instruments and the standard deviation against ACE-FTS (FmO time series between 60°S-30°S and 30°S-30°N are given in the supplement). For ClO, which is not retrieved in ACE-FTS v3.6, the standard deviation against SMILES

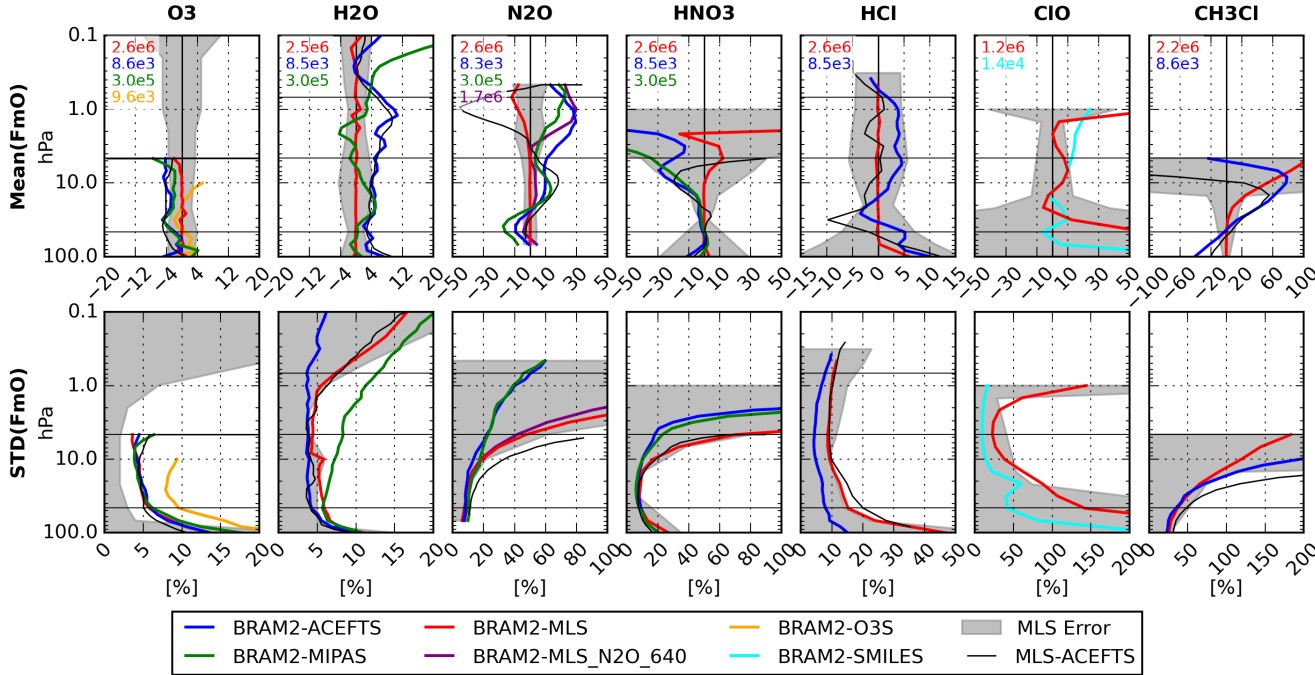

**Figure 5.** Mean (top row) and standard deviation (bottom row) of the differences between BRAM2 and observations from MLS (red lines), ACE-FTS (blue lines), MIPAS (green lines), MLS_N2O_640 (purple lines), NDACC ozonesondes (orange lines) and SMILES (cyan lines). The statistics are in %, normalized by the mean of BRAM2 and are taken between 30°N-60°N, 0.1-100 hPa and the 2005-2017 period. The statistics are calculated for, from left to right, $O_3$, $H_2O$, $N_2O$, $HNO_3$, HCl, daytime ClO and $CH_3Cl$. The approximate numbers of observed profiles used in the FmO statistics are given in the upper left corner of the top row plots using the instrument color code. The gray shaded area in the mean and standard deviation plots corresponds, respectively, to the MLS accuracy and precision, as provided in the MLS data quality document (see L2015). The thin black profiles represent the mean (top row) and standard deviations (bottom row) between MLS and ACE-FTS found in validation publications (see text for details). The horizontal black lines denote levels where time series are shown in Fig. 6.

is shown. The time series are in % except for ClO which is in ppbv. For this species, only daytime observations are taken into account in Fig. 5 and 6.

In general, BRAM2 represents a good proxy for MLS. The biases against MLS are smaller than the MLS accuracy so that they are not significant (Fig. 5). Moreover, the standard deviations against MLS and the MLS precision are usually in good agreement, except for $O_3$. Time series of the bias against MLS is in general very stable with negligible amplitude in the seasonal

5   variations, except for $N_2O$ at 0.68 hPa, $HNO_3$ at 4.6 hPa, ClO at 4.6 and 46 hPa and $CH_3Cl$ at 4.6 hPa (see Fig. 6).



The standard deviation against ACE-FTS is usually smaller than the standard deviation against MLS, which indicates that the variability in MLS observations is larger than that in ACE-FTS (Fig. 5). Also, the standard deviation against ACE-FTS is usually stable in time (Fig. 6).

The biases against ACE-FTS are in general similar to the differences between MLS and ACE-FTS calculated in published validation studies (see Sect. 4), except for HCl, $N_2O$ above 3 hPa, $HNO_3$ above 10 hPa and $CH_3Cl$ above 20 hPa. This means

that most of the differences between BRAM2 and the independent observations are due to the difference between these datasets and MLS. Also, the standard deviations against ACE-FTS are as good as or better than those from direct comparisons between MLS and ACE-FTS (except for $O_3$ below 40 hPa). This suggests that a significant part of the standard deviations of (MLS-ACEFTS) calculated in validation studies are due to sampling error introduced by the collocation approach. In our case, the sampling error is replaced by the representativeness error arising from the limited spatial and temporal resolution of the data

assimilation system. We thus conclude that the representativeness errors within BRAM2 are smaller than the sampling errors inherent in validation studies based on collocation of profiles.

Thus, in general, BRAM2 mean values and their variability agree well with the observations. Let us now discuss these statistics from species to species (see Sect. 5.5 for a summary of BRAM2 evaluation in the different regions):

**$O_3$:** We recall that ozone is not assimilated at altitudes higher (i.e. pressure lower) than 4 hPa due to a BASCOE model ozone

deficit (see Sect. 3.1) and comparisons above that level are not shown. Bias against MLS is around $-2\%$ at 5 hPa and negligible below (see Fig. 5). Biases against ACE-FTS and MIPAS are usually negative and around $-5\%$. Bias against ozonesondes is within $\pm4\%$. The standard deviations against MLS, ACE-FTS and MIPAS are similar, ranging from 4% at 10 hPa to $\sim15\%$ at 100 hPa. Against ozonesondes, the standard deviations are larger by around 5%, likely due to the higher vertical resolution of ozonesondes (100 m) against the model (1-3 km).

Time series of the bias against all instruments at 4.6 and 46 hPa are stable from year to year, with seasonal variations usually smaller than 5% except against MIPAS at 4.6 hPa ($<10\%$) (Fig. 6). The standard deviations against ACE-FTS are in general lower than 5% with small seasonal variations at 4.6 and 46 hPa. Similar statistics are found in



**Figure 6.** Time series of the monthly mean differences between BRAM2 and the different observational datasets in the northern hemisphere mid-latitudes (30°N-60°N) at three pressure levels (from left to right: 0.68, 4.6 and 46 hPa) and for (top to bottom) $O_3$, $H_2O$, $N_2O$, $HNO_3$, HCl, daytime ClO and $CH_3Cl$. Values are in % except for ClO, which is shown in ppbv. The gray shaded area represents the standard deviation of the differences between BRAM2 and ACE-FTS except for ClO, where SMILES data are used.





the southern hemispheric mid-latitudes and in the Tropics above 50 hPa (see Figs. S1-S4). Given the good agreement between BRAM2 and the observations, we recommend BRAM2 $O_3$ in the MS for scientific use between 4 and 100 hPa.

**$H_2O$:** The bias against MLS is negligible (Fig. 5). A positive bias lower than 10% is found against ACE-FTS. The agreement against MIPAS is also very good, within $\pm 5\%$ below 0.5 hPa. Between 1 and 50 hPa, the standard deviations against MLS, ACE-FTS and MIPAS are around 5, 4 and 8%, and remain smaller than 5% against ACE-FTS above 1 hPa. For MLS and MIPAS, the standard deviations increase up to 15% and 20% at 0.1 hPa, respectively

Biases are stable over time, with small seasonal variation against MLS (amplitude around <2%), ACE-FTS (<4%) and MIPAS (<8%) between 0.68 and 46 hPa (Fig. 6). Standard deviations against ACE-FTS are also stable (<5%), displaying negligible seasonal variations. As for ozone, similar statistics are found in the southern hemisphere and in the Tropics above 50 hPa (see Figs. S1-S4) and we recommend BRAM2 $H_2O$ for scientific use in these regions.

**$N_2O$:** At altitudes above 3 hPa, BRAM2 is poorly characterized by comparison against observations. The standard deviations against MLS are larger than 100% (Fig. 5) and the time series of the FmO are noisy, with peak to peak variations larger than 50%. In the upper stratosphere, the MLS precision degrades to around 65% at 2 hPa, which limits the constraint of the assimilated observations on the reanalysis.

At altitudes below 3 hPa, the BRAM2 bias against MLS is negligible (Fig. 5). The agreement with MLS_N2O_640, ACE-FTS and MIPAS is good, within $\pm 5\%$, $\pm 10\%$ and $\pm 15\%$, respectively. The standard deviations of the FmO are relatively small below 10 hPa (<10%) and increase above that level. At 3 hPa, values are $\sim 25\%$ against ACE-FTS and MIPAS, $\sim 60\%$ against MLS_N2O_640 and $\sim 80\%$ against MLS. This highlights the higher sensitivity (or lower random error) of ACE-FTS and MIPAS compared to MLS for upper stratospheric $N_2O$.

The bias time series are stable against MLS, with seasonal amplitude lower than 2% at 4.6 and 46 hPa (Fig. 6). Although larger and with greater seasonal variations, biases against independent data are relatively good, between [10,20]% at 4.6 hPa and [5,10]% at 46 hPa, depending on the instrument. Time series of the standard deviation against ACE-FTS also show seasonal variations with peak to peak amplitude around 20% at 4.6 hPa and $\sim 5\%$ at 46 hPa. One also notices a drift in the time series of the bias at 46 hPa between BRAM2 and the three independent datasets. Analyses of the

deseasonalized time series of the biases reveal a significant drift of -5, -7 and -5%/decade against ACE-FTS, MIPAS and MLS_N2O_640 for the period 2005-2012 and -10% against ACE-FTS for 2005-2017 (not shown). This drift has been mentioned in Froidevaux et al. (2019) and is under investigation by the MLS team (Livesey and colleagues, in prep). This result suggests that BRAM2 could be used as transfer functions between the instruments to correct for their drifts.

Similar agreement is found in other latitude regions (see Figs. S1-S4). Therefore at altitudes below 3 hPa and excluding trend analysis, we recommend BRAM2 $N_2O$ for scientific use in the middle stratosphere. At altitudes above 3 hPa, BRAM2 should not be used without consulting the BASCOE team.

**HNO$_3$:** As for $N_2O$, BRAM2 HNO$_3$ is poorly characterized by comparison against observations at altitudes above 3 hPa. The biases against the three instruments are large and disagree in sign and size (Fig. 5). Above that level, MLS precision degrades (reaching 0.6 ppbv i.e. 6 times the typical amount of HNO$_3$ at that level) and the constraint by the assimilated observations on BASCOE is weak.

At altitudes below 3 hPa, BRAM2 agrees well with MLS (within [0,10]%, see Fig. 5). Below 10 hPa, the agreement against ACE-FTS and MIPAS is between [-10,1]% for both instruments. A large negative bias against ACE-FTS and MIPAS is found between 3-10 hPa, within [-10,-30]% against ACE-FTS and up to $-50$% against MIPAS. The standard deviations against all instruments are minimal around 30 hPa ($<$10%) and increase at higher and lower levels. Against ACE-FTS and MIPAS, the values remain $<$20% between 3-100 hPa.

These values are stable over time at 4.6 and 46 hPa, while displaying significant seasonal oscillations at 4.6 hPa, around 20% against all instruments (Fig. 6). At 46 hPa, the seasonal variations against MLS are negligible, and are around 10% and 5% against, respectively, ACE-FTS and MIPAS. The standard deviation against ACE-FTS is small and stable at that level (around 5%). At 4.6 hPa, the standard deviation is larger with greater seasonal oscillation, from $\sim$10% during summer to $\sim$20% during winter, likely due to the polar influence during the winter.

Similar statistics are found in the southern hemisphere and in the Tropics (see Figs. S1-S4) and we recommend BRAM2 HNO$_3$ for scientific use between 3 and 100 hPa. The use of BRAM2 HNO$_3$ above 3 hPa should be done in consultation with the BASCOE team.

**HCl:** As shown in Fig. 5, mean differences against MLS are negligible between 0.4 and 70 hPa (0.4-50 hPa in the Tropics, see the Fig. S2) and increase to 5% at 100 hPa (30% in the Tropics). There is good agreement with ACE-FTS, within ±5% between 0.4-70 hPa (50 hPa in the Tropics) with a bias increasing downward to 10% at 100 hPa (> 50% in the Tropics). The difference (BRAM2-ACEFTS) is larger by around 5% than the difference (MLS-ACEFTS) from Froidevaux et al. (2008). This is due to the different version of MLS and ACE-FTS used here (v4 and v3.6, respectively) and in Froidevaux et al. (v2 and v2.2), and the fact that the HCl amount has been reduced by around 5% in the latest version of ACE-FTS data (Waymark et al., 2013). Our comparison is thus an update to Froidevaux et al. (2008). The standard deviation against ACE-FTS is around ∼5% at 5 hPa and increases at higher and lower altitude to around 10%.

Bias time series are very stable against MLS with negligible seasonal variations (Fig. 6). On the other hand, a small drift is noticeable in the bias time series against ACE-FTS at 0.68 and 4.6 hPa. The MLS HCl v4 standard product assimilated for BRAM2 is retrieved from the band 14 of the 640 GHz radiometer, while the band 13 – more sensitive to HCl – was originally planned. This change of strategy by the MLS retrieval team was due to the deterioration of the band 13 which was turned off in 2006 (see L2015). For this reason, the MLS HCl from band 14 (and BRAM2) are not suited for detailed trend studies in the USLM. Again, this suggests the possibility of using BRAM2 as a transfer function between the instruments to correct for their relative drifts. At 46 hPa, no significant drift is found and the bias against ACE-FTS displays a 10% peak to peak variation.

Based on these comparisons, we recommend BRAM2 HCl for scientific use between 0.4 and 100 hPa (50 hPa in the Tropics), but it cannot be used for trend studies.

**ClO:** For ClO, the analysis increments are very small in the middle stratosphere and the bias (BRAM2-MLS) and (CTRL-MLS) are similar, as suggested by Fig. 3. The bias against MLS is within ±10% between 1.5 and 30 hPa (see Fig. 5). The standard deviation is minimal (∼25%) around 4 hPa where ClO abundances are maximum and increases to around 50% at 1.5 and 15 hPa. Mean differences against SMILES agree well with those against MLS between 1.5 and 30 hPa. The standard deviations against SMILES are <20% in this altitude range, which is much smaller than against

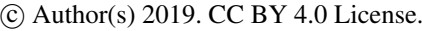

MLS, suggesting higher precision of SMILES compared to MLS. Comparison against MIPAS ClO is not shown because MIPAS V5_ClO_22[0_or_1] is only valid under conditions of chlorine activation in the polar winter (see Sect. 2.2.2).

Time series of the bias against MLS show small seasonal variations (<0.04 ppbv) at 4.6 and 46 hPa (Fig. 6). Similar values are found in the southern mid-latitudes and in the Tropics (see Figs. S1-S4). We conclude that BRAM2 ClO in the middle stratosphere is more a CTM product than a data assimilation product. Nevertheless, BRAM2 ClO in the middle stratosphere can be recommended for scientific use between 1 and 70 hPa and should be used in consultation with the BASCOE team outside this vertical range.

**CH$_3$Cl:** Below 30 hPa, the BRAM2 bias against MLS is very small (<5%) and increases upward to ∼60% at 10 hPa (Fig. 5). The bias against ACE-FTS is larger, from around −40% at 100 hPa to ∼70% at 10 hPa. The standard deviations are ∼25% at 100 hPa and increase to ∼50% at 30 hPa against both instruments. At 10 hPa, the standard deviations are larger than 100% for both instruments.

At 46 hPa, the time series of the biases display negligible seasonal variations against MLS (within ±5%) and small seasonal variations against ACE-FTS (<15%, see Fig. 6). The agreement is better in the Tropics where, below 10 hPa, the bias is lower than 5% against MLS and within ±20% against ACE-FTS (see Fig. S2 and S4).

Above 30 hPa at mid-latitudes (10 hPa in the Tropics), the agreement between BRAM2 and MLS degrades. More worrying is that CTRL agrees better with MLS observations than BRAM2 (see Fig. 3), indicating that MLS observations are not properly assimilated. The reason for this issue is probably twofold. First, there is a relatively large number of negative MLS CH$_3$Cl observations above 10 hPa, and second, the MLS averaging kernels are not used in the BASCOE observations operator. Assimilating negative data is not an issue as long as the overall analysis is positive which is not always the case with CH$_3$Cl. In BASCOE, negative analyses are clipped to nearly zero ($10^{-25}$) which in the case of CH$_3$Cl introduces a positive bias in the analysis. Since significant information in the retrieved profiles comes from the a-priori above 10 hPa (see L2015, their Fig.3.3.2), the use of the averaging kernels would help to ensure positiveness of the analysis. Unfortunately, this issue was not considered before starting the production of BRAM2. Consequently, we recommend the use of BRAM2 CH$_3$Cl only below 30 hPa at mid-latitudes and 10 hPa in the Tropics.

## 5.2 Lower Stratospheric Polar Vortex (LSPV)

Figure 7 shows the evolution of the southern hemispheric (SH) polar vortex composition from MLS observations, BRAM2 and CTRL in 2009. Values correspond to daily means in the inner vortex, i.e. between 90°S-75°S of equivalent latitude. The vertical domain is between 320 to 700 K potential temperature, approximately between 10 and 100 hPa. The SH inner polar vortex was chosen because it is the region where CTRL differs most from the reanalysis. Evaluating BRAM2 in these conditions is thus a
stronger test for the quality of the reanalysis. The species shown in Fig. 7 and discussed throughout this section are $O_3$, $H_2O$, $N_2O$, $HNO_3$, HCl and ClO. For ClO, only daytime values are taken into account.

Qualitatively, there is a very good agreement between BRAM2 and MLS, as expected, for the patterns associated with both chemical and dynamical processes. For the chemistry, the loss of HCl by heterogeneous chemistry and the activation of ClO are well reproduced by BRAM2, as is the loss of $HNO_3$ and $H_2O$ by denitrification and dehydration. The ozone depletion
in BRAM2 that occurs in September-October is also in very good agreement with MLS. Dynamical patterns are also in good agreement. The descent of air from above 700 K that starts in May and ends in October, exhibited by the decrease of $N_2O$ and the increase of $H_2O$ and $O_3$, is well reproduced by BRAM2. Dynamical patterns of shorter timescales are also well reproduced by BRAM2, e.g. the increase of $N_2O$ in late July and late August.

Comparison between the CTRL and BRAM2 shows the regions where MLS observations correct the bias in the BASCOE
CTM. For chemical patterns, the loss of HCl is relatively well represented in the model, between 450 and 650 K. Below 450 K, modeled HCl overestimates the observations. Above 400 K, the loss of $HNO_3$ is also well reproduced by the model while the model has a negative bias below that level. The model also slightly underestimates the ClO activation and the loss of $H_2O$ by dehydration. The good performance of CTRL, especially for HCl, contrasts with recent studies showing the difficulties of CTMs (Lagrangian and Eulerian) to simulate the loss of HCl by heterogeneous chemistry (Wohltmann et al., 2017; Grooß
et al., 2018). Note that the BASCOE CTM is based on a relatively simple PSC parameterization and that its parameters have been tuned to improve the model representation (see Sect. 3.1). In other words, it does not include the state of the art of heterogeneous chemistry treatment as in these other studies and this setup is justified to create a reanalysis in good agreement with observations.



**Figure 7.** Time series of (from top to bottom) daily mean MLS profiles, the corresponding values of BRAM2, the corresponding values of CTRL, the mean difference between BRAM2 and MLS and the associated standard deviation. Values are shown between May-November 2009 in the lower stratospheric inner vortex (i.e. within 90°S-75°S equivalent latitude and within 320-700 K) for (from left to right) O₃, H₂O, N₂O, HNO₃, HCl and ClO. Only daytime values of ClO are considered in the mean calculations. White areas correspond to locations/dates without valid observations.

For dynamical patterns, CTRL shows a more pronounced bias. Descent of air (exhibited by high values of $H_2O$ and low values of $N_2O$ between 600-700 K), which is correctly reproduced from the beginning of the simulation, is abruptly interrupted in July, probably due to a weakening of the polar vortex at that time. This bias can be attributed to the coarse horizontal resolution of the model (Strahan and Polansky, 2006) and is successfully corrected in BRAM2.

Figure 7 also displays the daily mean and standard deviation of the differences between BRAM2 and MLS. The differences

are normalized by the daily mean of BRAM2. In general, large biases (around 50%) correspond to conditions with very low absolute values, i.e. for (1) $HNO_3$ and HCl between July and Oct and below 650K, (2) ClO outside conditions of chlorine activation and (3) $N_2O$ between 600-700 K during the descent of upper stratospheric air. Large bias also occurs for $H_2O$ below 400 K, i.e. in the UTLS. When chlorine is activated (i.e. when ClO abundance is greater than 1 ppbv), the mean differences between BRAM2 and MLS are below 10%, well within the MLS accuracy (0.1 ppbv, see L2015). Bias also increases for $O_3$

in late September, during the developement of the ozone hole, but to a reasonable extent (10%).

The standard deviations of the differences also increase when the concentration of the species is very low. In particular, the standard deviation can be higher than 100% for $N_2O$, $HNO_3$ and HCl in the above mentioned regions. In these cases, standard deviations are more relevant when unnormalized (i.e. in vmr units) and the corresponding values for these three species are, respectively, 10, 0.2 and 0.1 ppbv (not shown).The standard deviation for $O_3$ also increases and is maximum (between 25-50%)

in late September between 400-500K.

Comparison of BRAM2 in LSPV conditions with independent observations and for other years than 2009 is done in Fig. 8. It shows time series of monthly FmO between 90°S-75°S of equivalent latitude for the 2005-2017 period and at two potential temperature levels: 650 and 450 K (~15 and ~50 hPa, similar figures for the outer vortex and the Arctic winters are shown in the supplement). Statistics shown are the bias against the different instruments and the standard deviation against ACE-FTS.

For ClO, which is not retrieved in ACE-FTS v3.6, the standard deviation against MIPAS is shown. The time series are in % except for ClO which is shown in ppbv. For this species, only daytime observations are taken into account. As expected, comparisons against MLS provide lower biases than against independent observations. Let us discuss Fig. 8 for each species individually:

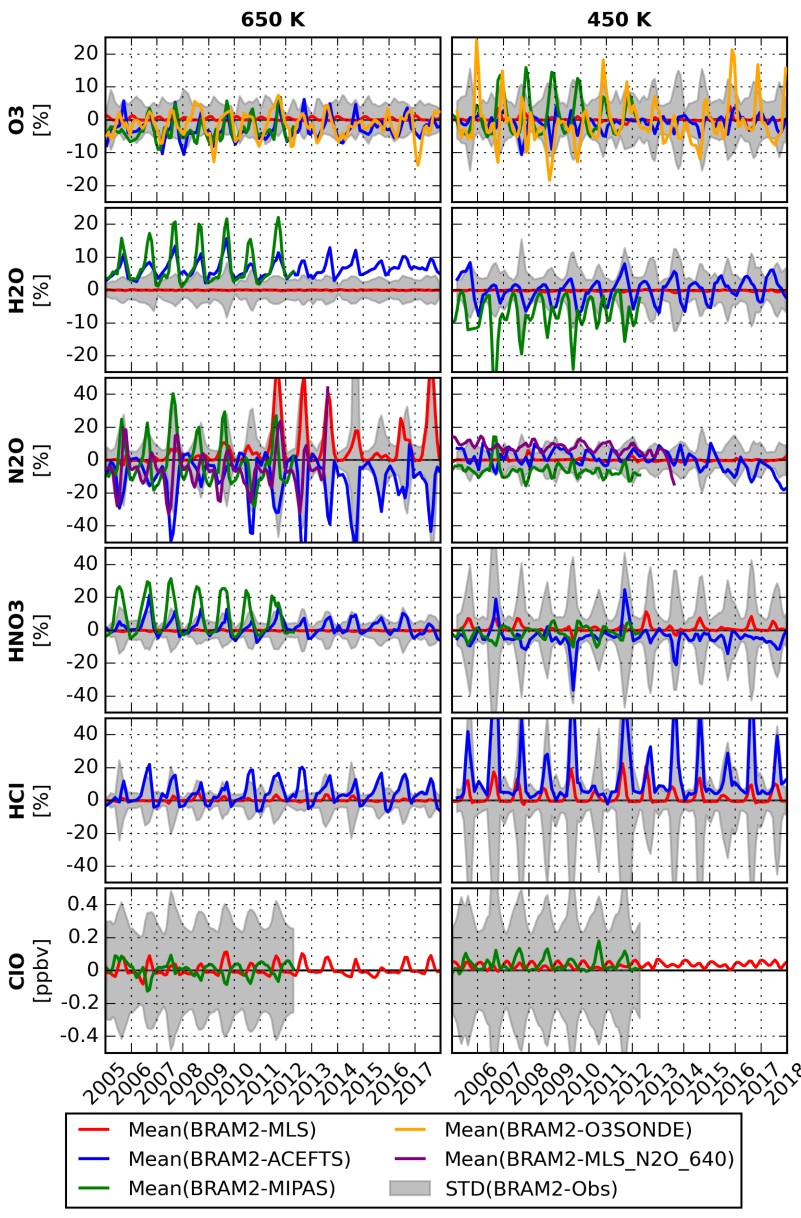

**Figure 8.** Time series of the monthly mean differences between BRAM2 and the different observational datasets between 90°S-75°S of equivalent latitude at two potential temperature levels (from left to right: 650 and 450 K) and for (top to bottom) $O_3$, $H_2O$, $N_2O$, $HNO_3$, HCl and daytime ClO. Values are in % except for ClO which is shown in ppbv. The gray shaded area represents the standard deviation of the differences between BRAM2 and ACE-FTS except for ClO where MIPAS data are used.





**O$_3$:** BRAM2 and MLS agree very well, their mean differences are below 2%. The agreement against independent obser-
vations is good, around -5% at 650 K and ±5% at 450 K, against ACE-FTS and MIPAS. Larger differences occur
against ozonesondes, likely due to the interpolation of the high resolution ozonesonde profiles to potential temperature.
Nevertheless, the agreement with ozonesondes is generally better than ±10%. Compared to intercomparison between
instrument climatologies, done in SPARC/IO3C/GAW (2019, Fig. 4.1.19 and 4.1.20), the comparison of BRAM2 against
5       independent data displays lower biases. The standard deviations against ACE-FTS are generally smaller than 10% with
larger values (around 15%) in the lower stratosphere (450K) during Antarctic springs. We recommend the use of BRAM2
O$_3$ for scientific use in the LSPV.

**H$_2$O:** BRAM2 and MLS agree very well, the biases are negligible (<1%). The mean bias against ACE-FTS is similar to
the value found in the middle stratosphere, although with larger seasonal variations. The standard deviations against
10      ACE-FTS are very small: from 10% during dehydration periods to 5% otherwise. The biases against MIPAS are larger
and also have larger seasonal variations, in particular at 650 K where the descent of upper atmospheric air is significant.
In cold conditions (i.e. during polar winter at 650K), the MIPAS averaging kernels (AK) are smoother than in warm
conditions (i.e. during polar summer). These AK were applied to a few BRAM2 ModAtObs profiles at MIPAS during
summer and winter polar conditions, resulting in a better agreement with MIPAS (not shown). We recommend the use
15      of BRAM2 H$_2$O for scientific use in the lower stratospheric polar vortex.

**N$_2$O:** The amount of N$_2$O decreases during polar winter, in particular at 650 K in the beginning of Antarctic spring. While
the normalized differences against MLS and the independent observations can be greater than 50%, the unnormalized
differences are always <20 ppbv and typically <10 ppbv, which is small given the relatively low amount of N$_2$O (see
typical values in Fig. 7). At 650 K, the standard deviation against ACE-FTS displays larger seasonal variations (up to
20      20-50% in early spring). At 450 K, where the abundance of N$_2$O is larger, the mean biases are generally within ±5%
against MLS and ±15% against the independent datasets. As discussed in Sect. 5.1, one can see a drift between BRAM2
and the independent datasets, likely due to a drift in the MLS N$_2$O standard product. The standard deviation against



ACE-FTS is stable, displaying a small seasonal variation, typically between [5,15]%. Overall, BRAM2 $N_2O$ is reliable in the LSPV and is recommended for scientific use except for trend studies.

**HNO$_3$:** At 650 K, BRAM2 agrees very well with MLS (<2% bias). Biases against ACE-FTS and MIPAS are small, usually within [0,15] and [0,25]%, respectively. These positive biases against ACE-FTS and MIPAS are of opposite sign compared to those in the middle stratosphere, in agreement with observation intercomparison (SPARC/IO3C/GAW, 2019, their Fig. 4.13.3). The standard deviation against ACE-FTS is small, within [3,10]% depending on the season.

At 450 K, where $HNO_3$ is lost by PSC uptake and denitrification, BRAM2 overestimates MLS during late winter/early spring by around 5%, which is within the MLS accuracy. Biases against ACE-FTS and MIPAS oscillate around $-5\%$. The larger deviations against ACE-FTS are likely due to the sparse sampling of this instrument. The standard deviation of the differences against ACE-FTS is larger during the denitrification period (reaching 20 to 50% depending on the year). This means that data assimilation can correct the model bias due to the BASCOE PSC parameterization but does little to improve its lack of precision. Overall, BRAM2 $HNO_3$ in LSPV is a reliable product and is recommended for scientific use even though it is affected by a large positive relative bias (but small when unnormalized) in regions completely denitrified by PSC sedimentation.

**HCl:** Similar conclusions hold for HCl. BRAM2 agrees well with MLS and ACE-FTS at 650 K, at the upper limit of PSC activity (see Fig. 7). The bias against MLS is better than 5%, within the MLS accuracy. Against ACE-FTS, the bias is usually positive (as in mid-latitudes) with seasonal variations between $-5$ and $+20\%$, which is satisfactory considering the very large loss of HCl by heterogeneous reactions with PSCs. The situation is similar at 450 K. Even though biases are larger (up to $\sim$20 and $\sim$50% against MLS and ACE-FTS), the unnormalized biases remain smaller than 0.2 ppbv (not shown). Standard deviations against ACE-FTS are large during winter time at 450 K (up to 50%). The unnormalized standard deviation is around 0.2 ppbv. Again, as for $HNO_3$, BRAM2 HCl in the LSPV is a reliable product and is recommended for scientific use even though it is affected by large positive relative bias (but small when unnormalized) in regions where HCl has been completely destroyed by heterogeneous reactions on the surface of PSCs.



**ClO :** At altitudes where ClO production by chlorine activation is high (at 450 K in Fig. 8), the biases against MLS and MIPAS are positive, within [0,0.05] and [0,0.1] ppbv, respectively. Standard deviation against MIPAS can be as large as 0.5 ppbv during chlorine activation conditions (around 100%). Comparison of BRAM2 with SMILES ClO has been done for the chlorine activation period in the Arctic winter 2009-2010 (not shown). Around 500 K (i.e. the level where ClO reaches a maximum during chlorine activation), BRAM2 overestimates SMILES by around 10% with a standard deviation around

50%. On average, BRAM2 agrees well with MIPAS and SMILES observations in the LSPV but displays large variability in the comparison. Again, BRAM2 ClO in the LSPV is a reliable product when ClO is enhanced by chlorine activation and is recommended for scientific use in these conditions.

## 5.3    Upper Stratosphere lower mesosphere Polar Vortex (USPV)

Upper stratosphere lower mesosphere polar winters are influenced by the descent of mesospheric and thermospheric air into

the stratosphere, in particular by the descent of $NO_x$ (i.e. $NO+NO_2$), CO and $H_2O$ (Lahoz et al., 1996; Funke et al., 2005, 2009). Enhanced stratospheric $NO_x$ induces production of $HNO_3$ by ion cluster chemistry (e.g., Kvissel et al., 2012). In the Arctic, all these processes may be affected by stratospheric major warmings that displace or split the polar vortex (Charlton and Polvani, 2007). The BASCOE CTM does not account for mesospheric or thermospheric sources, nor for ion chemistry. On the other hand, Lahoz et al. (2011) have shown that the BASCOE system constrained by MLS $H_2O$ observations was able to

describe the Arctic vortex split of January 2009. In this section, we will evaluate how the results of Lahoz et al. (2011) could be extended to BRAM2 for other years than 2009, in both hemispheres and for $N_2O$, $HNO_3$ and CO.

Note that CTRL is not shown in this section. It displays large disagreement with MLS and/or BRAM2 which only highlights the CTM limitations explained above.

Figure 9 shows the time series of MLS and BRAM2 during the USPV Arctic winter 2016-2017, between 0.1 and 10 hPa

and between 60°N-90°N. Note that the figure is not given in the equivalent latitude/theta view as in the LSPV in order to keep the upper model levels in the discussion. This winter was subject to intense dynamical activity with two strong warmings – although not major – where the vortex almost split at the end of January and February (not shown). Around these dates, discontinuities in the time series of MLS $H_2O$ and CO are clearly visible, and well reproduced by BRAM2. Time series of



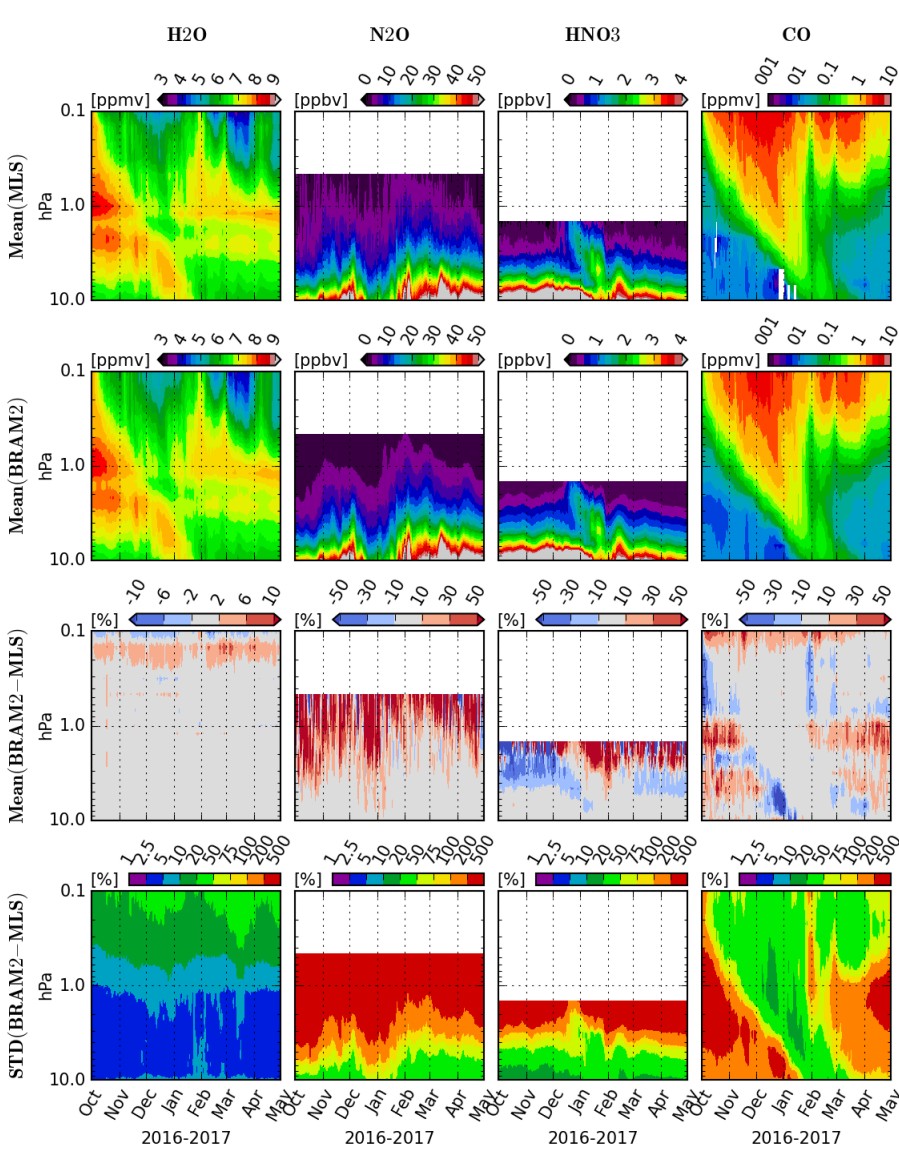

**Figure 9.** Time series of (from top to bottom) daily mean MLS profiles, the corresponding BRAM2 values, the mean differences between BRAM2 and MLS and the associated standard deviation. Values are shown between October 2016 and May 2017, between 90°N-60°N and 0.1-10 hPa for (from left to right) $H_2O$, $N_2O$, $HNO_3$ and CO.

MLS $HNO_3$ show enhanced values in January around 3 hPa most likely due to ion chemistry. While this process is not included

in the BASCOE CTM, BRAM2 agrees well with MLS. The average BRAM2 $N_2O$ also agrees well with MLS below 1 hPa.

Figure 9 also displays the mean and standard deviation of (BRAM2-MLS). As for previous comparisons shown in this paper,

the mean and standard deviation are large when the abundance of the species is relatively low, i.e. in the upper parts of the plots

for $N_2O$ and $HNO_3$, or for CO outside the period of enhancement by descent of mesospheric air. Otherwise, the agreement is

relatively good for CO during descent of mesospheric air, during production of $HNO_3$ by ion chemistry, or at lower altitude

for $N_2O$ when its abundance exceeds ∼20 ppbv. In those cases, the bias is around ±10%, while the standard deviations of the

differences are relatively large but still acceptable (<50%). For $H_2O$ the agreement is very good below 0.2 hPa, where bias

and standard deviation are usually lower than ±2% and 20%, respectively.

Figure 10 shows the FmO statistics between BRAM2 and observations from MLS, ACE-FTS, MIPAS and MLS_N2O_640

in the USPV. These statistics are calculated between 60°N-90°N and 0.1-10 hPa for the January-February 2005-2017 periods,

these months being observed by ACE-FTS (for MIPAS and MLS_N2O_640, the datasets end in 2012 and 2013, respectively).

The figure also shows the MLS accuracy and precision, and the mean and standard deviation of the differences (MLS-ACEFTS)

estimated by S2017. A similar figure for the southern polar winter is provided in the supplement (Fig. S8).

The FmO statistics are similar to those found in the middle stratosphere (see. Sect. 5.1) for $H_2O$ and $N_2O$. FmO statistics

for $HNO_3$ are also somewhat similar to those found in the middle stratosphere. The major difference is the smaller bias found

between BRAM2 and MLS (<5%) at altitude below 3 hPa, approximately the upper level where $HNO_3$ is produced by ion

chemistry, well within the MLS accuracy. The FmO statistics of these three species are also stable from year to year (not

shown) and similar values are found in the southern hemisphere (see Fig. S8).

For CO, bias against MLS is small (<±5%) and well within the MLS accuracy. The biases against ACE-FTS or MIPAS are

similar, usually within ±10% with a maximum positive bias of +15% at 1 hPa. The bias against ACE-FTS agrees well with

the direct comparison between MLS and ACE-FTS which suggests that the BRAM2 bias comes from the differences between

the two instruments.

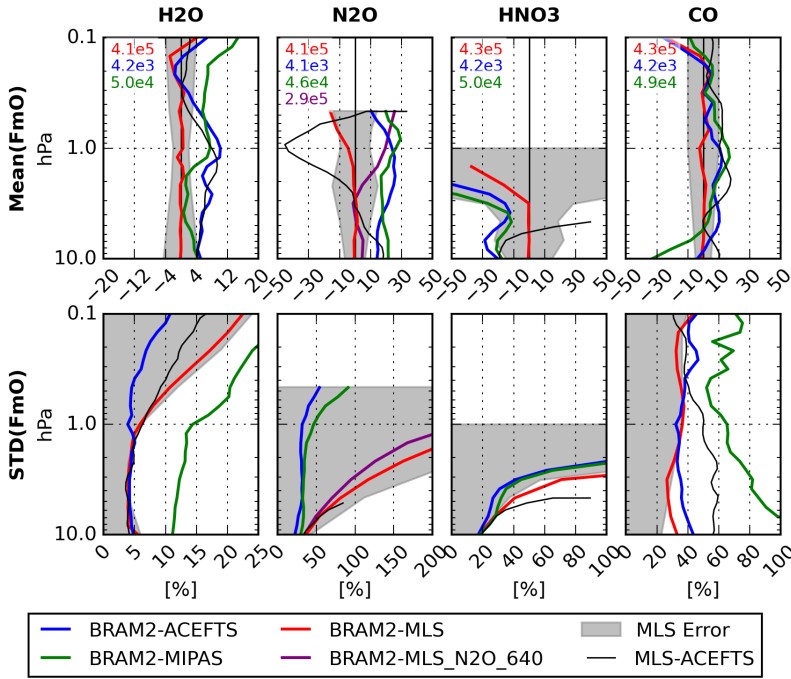

**Figure 10.** Mean (top row) and standard deviation (bottom row) of the differences between BRAM2 and observations from MLS (red lines), ACE-FTS (blue lines), MIPAS (green lines) and MLS_N2O_640 (purple lines). The statistics are in %, normalized by the mean of BRAM2 and are taken between 60°N-90°N, 0.1-10 hPa and the months January-February (i.e. during months observed by ACE-FTS) of the period 2005-2017. The statistics are calculated for, from left to right, $H_2O$, $N_2O$, $HNO_3$ and CO. The approximate numbers of observed profiles used in the FmO statistics are given in the upper left corner of the top row plots using the instrument color code. The gray shaded area in the mean and standard deviation plots corresponds, respectively, to the MLS accuracy and precision, as provided in the MLS data quality document (see L2015). The thin black profiles represent the mean (top row) and standard deviations (bottom row) between MLS and ACE-FTS found in validation publications (see text for details).

The standard deviations against MLS (∼35%) agree well with the MLS precision. BRAM2 provides similar deviations against ACE-FTS which are significantly lower than in the direct comparison between MLS and ACE-FTS. Against MIPAS, the standard deviation is relatively large, between 50 and 80%.

In the southern hemisphere, the biases against ACE-FTS and MIPAS are slightly larger by ∼5 and ∼10%, respectively (see Fig. S8). On the other hand, the standard deviations are smaller by around 10% against all datasets, with very good agreement

5    with the difference (MLS-ACEFTS). These statistics are relatively stable over the years. BRAM2 CO in the USPV agrees well with observations and is recommended for scientific use.

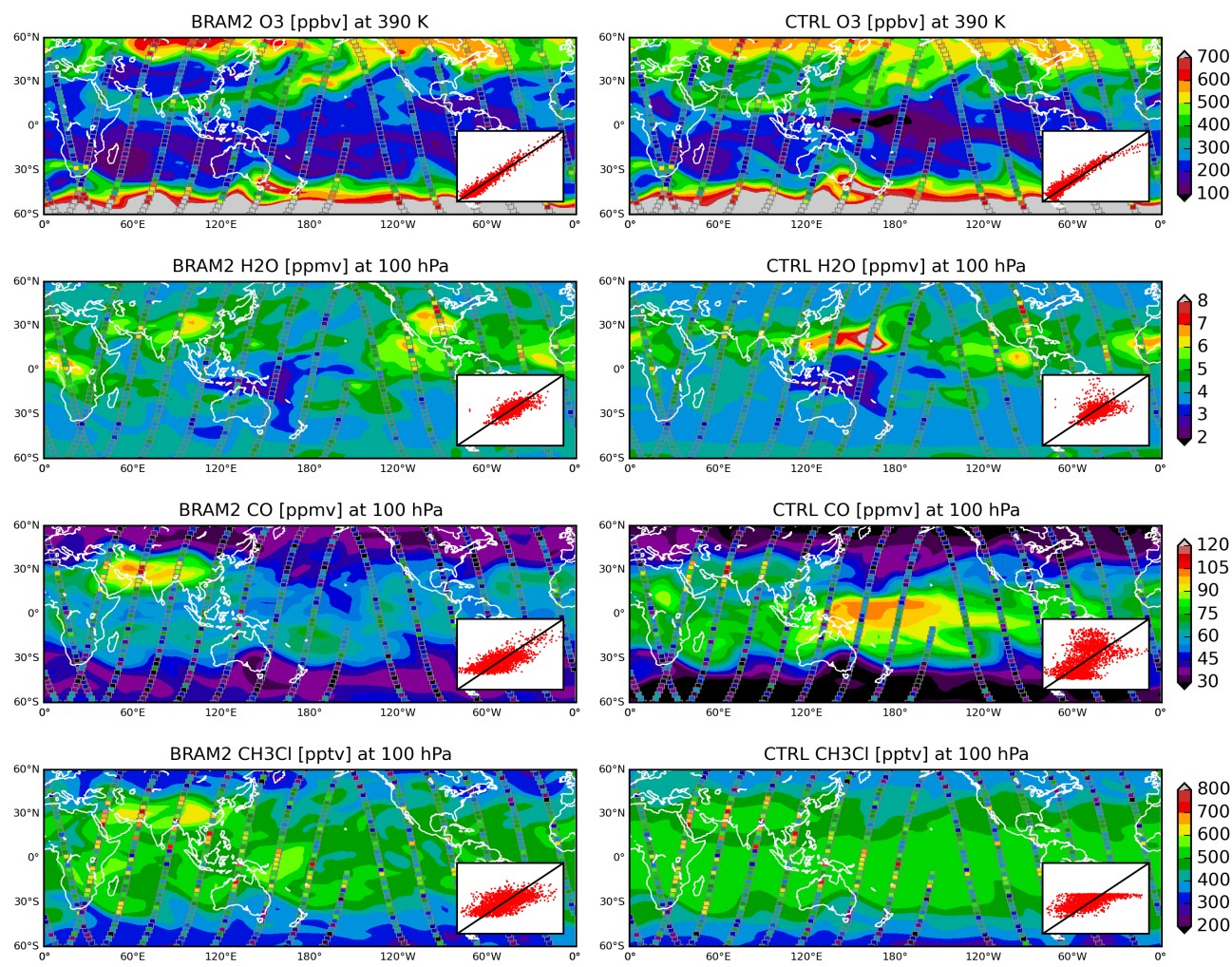

**Figure 11.** The filled contour maps (between 60°S-60°N) show the BRAM2 (left column) and CTRL (right column) distribution of, from top to bottom, O₃, H₂O, CO and CH₃Cl on July 14, 2009 at 12 UT at 390 K (O₃) or 100 hPa (H₂O, CO and CH₃Cl). Colored squares correspond to the MLS values between 6 and 18 UT on that date at the same levels. To improve the readability, only one in two MLS observations is shown. Scatter plots in the lower right corner of each map show the correlation between MLS and BASCOE (BRAM2 or CTRL, MLS on the x-axis, BASCOE on the y-axis) where all MLS data for that day are used, where BASCOE values are taken from the ModAtObs files (see Sect. 3.4) and where the black lines show the perfect correlation.

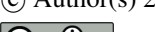



## 5.4 Tropical Tropopause Layer (TTL)

In the TTL, the evaluation of BRAM2 must take into account several limitations of the BASCOE CTM and the satellite observations. The BASCOE CTM does not include tropospheric processes, in particular the convection which is necessary to represent correctly vertical transport from the lower to the upper troposphere (Pickering et al., 1996; Folkins et al., 2002). Moreover, the BASCOE spatial resolution used for BRAM2 is relatively coarse to represent vertical and horizontal gradients in this region. Additionally, satellite observations are less reliable in the UTLS because large dynamical variability and steep gradients across the tropopause limit instruments with low temporal (occultation sounders such as ACEFTS) or vertical (emission sounders such as MLS and MIPAS) resolution. Also, cloud interference and saturation of the measured radiances pose challenges to the instruments, depending on the measurement mode applied (SPARC/IO3C/GAW, 2019).

In this section, the following BRAM2 species will be evaluated: $O_3$, $H_2O$, CO and $CH_3Cl$. Figure 11 shows the horizontal distribution of these species in the lower stratosphere from BRAM2 and CTRL on July 14, 2009 at 12 UT. To highlight the added value of the assimilation, MLS data between 6 and 18 UT on the same date are overplotted on each map. Finally, the qualitative agreement between MLS and the BASCOE values – BRAM2 or CTRL – corresponding to the selected situation is plotted on the lower right corner of each map. To highlight the differences between BRAM2 and CTRL, $O_3$ is shown at 390K (~80 hPa in the Tropics, ~150 hPa in mid-latitudes) while the other species are shown at 100 hPa. Despite the BASCOE model limitations, BRAM2 and MLS are in good agreement for $O_3$ (also confirmed by the high correlation between MLS and BRAM2 shown in the figure). For $H_2O$, $CH_3Cl$ and CO, the agreement is also generally good although the correlation between MLS and BRAM2 is less compact compared to $O_3$.

During boreal summer, the lower stratosphere is influenced by the anticyclonic circulation located above Asia which is associated with the Asian summer monsoon (e.g., Randel and Jensen, 2013). In Fig. 11, the anticyclone is marked by low $O_3$ abundance and high abundances of $H_2O$, CO and $CH_3Cl$ above Asia, indicating air of tropospheric origin. In contrast to CTRL, BRAM2 agrees relatively well with MLS in this region, which is not the case for CTRL. Also related to the Asian summer monsoon anticyclone is transport from mid-latitudes to the Tropics. This transport is marked by the $O_3$ tongue (values ~400 ppbv) starting in the northern east Pacific and ending above India, which denotes air of stratospheric origin (Randel and

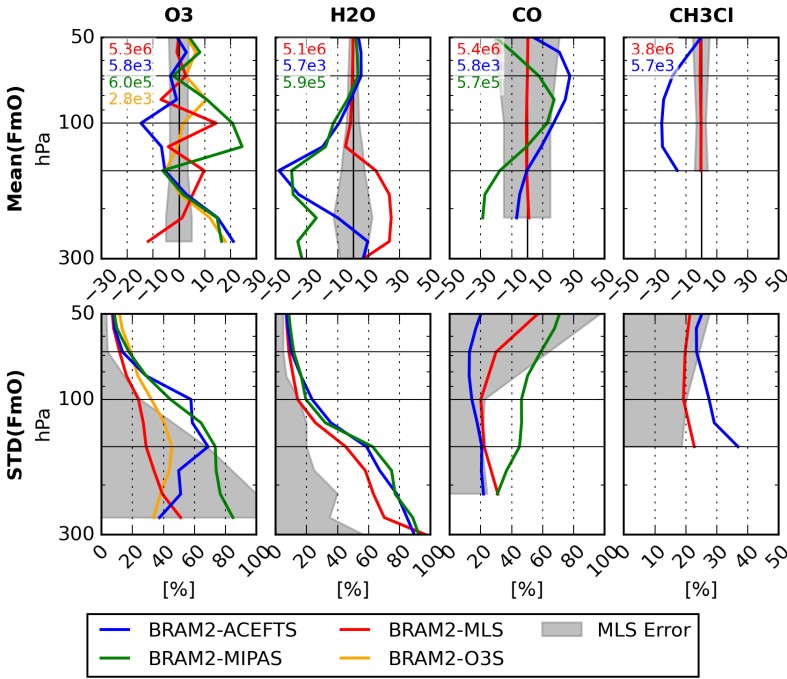

**Figure 12.** Mean (top row) and standard deviation (bottom row) of the differences between BRAM2 and observations from MLS (red lines), ACE-FTS (blue lines), MIPAS (green lines) and NDACC ozonesondes (orange lines). The statistics are in %, normalized by the mean of BRAM2 and are taken between 30°S-30°N, 50-300 hPa and the 2005-2017 period. The statistics are calculated for, from left to right, $O_3$, $H_2O$, CO and $CH_3Cl$. The approximate numbers of observed profiles used in the FmO statistics are given in the upper left corner of the top row plots using the instrument color code. The gray shaded area in the mean and standard deviation plots corresponds, respectively, to the MLS accuracy and precision, as provided in the MLS data quality document (see L2015). The horizontal black lines denote levels where time series are shown in Fig. 13.

Jensen, 2013, their Fig. 4, although for a different year). Even though this tongue is present in CTRL, BRAM2 and MLS show better agreement. The CTRL simulation of $H_2O$ and CO delivers unrealistically large abundances over the Western Pacific in the Tropics. The assimilation of MLS in BRAM2 removes these spurious features.

Figure 12 shows the FmO statistics between BRAM2 and observations from MLS, ACE-FTS, MIPAS and ozonesondes in the TTL for the 2005-2017 period. The figure also shows the MLS error budget (accuracy and precision). The differences
5  between MLS and ACE-FTS derived from validation papers (S2017 for $O_3$ and $H_2O$, Santee et al. (2013) for $CH_3Cl$) are not shown as it is the case in the MS and USPV sections, due to the lack of coincident profiles between the two instruments in the Tropics. Figure 12 is complemented by Fig. 13 showing time series of the mean differences of the FmO and the standard deviation against ACE-FTS or ozonesondes (for $O_3$). The results for each species are discussed individually:





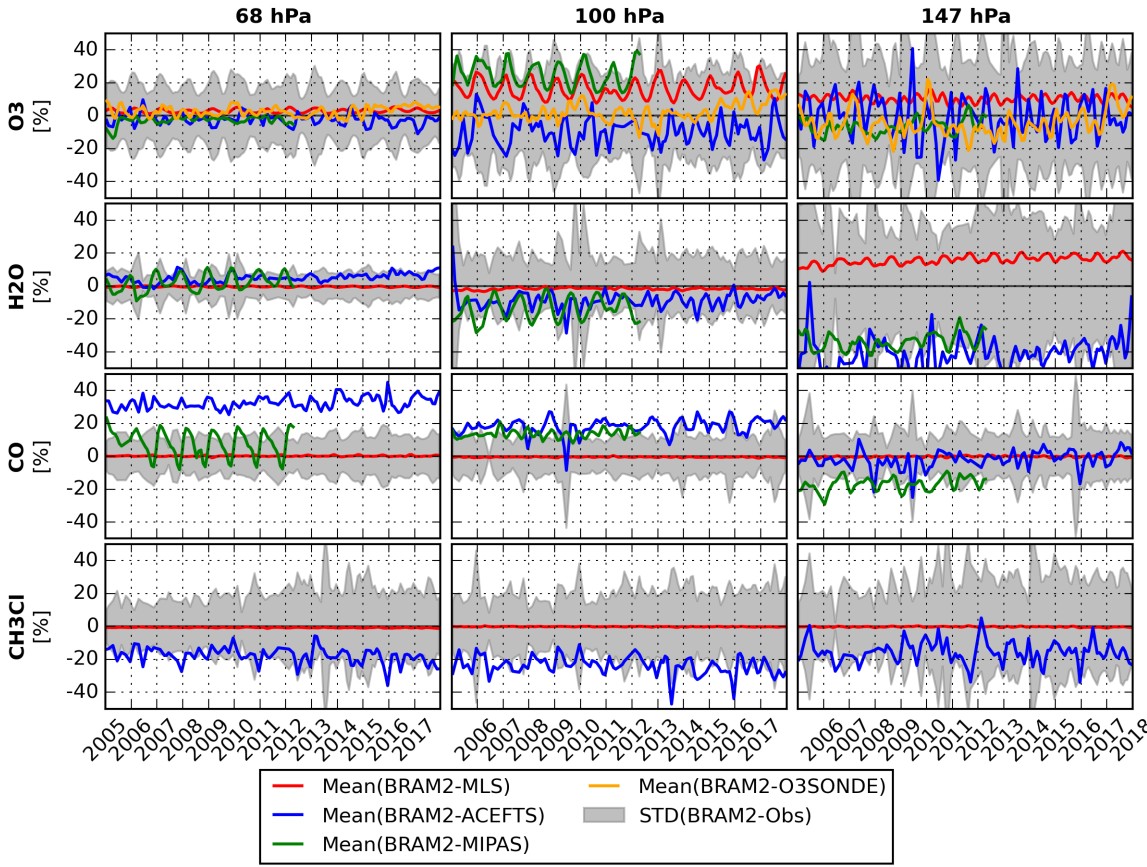

**Figure 13.** Time series of the monthly mean differences between BRAM2 and the different observational datasets in the TTL (30°S-30°N) at three pressure levels (from left to right: 68, 100 and 147 hPa) and for (top to bottom) $O_3$, $H_2O$, CO and $CH_3Cl$. Values are in %. The gray shaded area represents the standard deviation of the differences between BRAM2 and ACE-FTS except for $O_3$ where ozonesonde data are used.

$O_3$: The bias profile against MLS oscillates with a maximum of 15% at 100 hPa (Fig. 12) which is due to vertical oscillations in the MLS profiles (Yan et al., 2016, see L2015). The vertical resolution of BASCOE and MLS being similar in the TTL for $O_3$, the system cannot find a state that simultaneously minimizes the difference against all values of the MLS profiles, and it delivers a vertically smoother reanalysis. Because of these oscillations, the bias against MLS is larger than the MLS accuracy. Nevertheless, the agreement against ACE-FTS is good between 50 and 180 hPa, usually within 5%, except for a negative bias (−12%) at 100 hPa. The agreement against MIPAS is less good, with a 25% bias at 120 hPa. Note that satellite observations of $O_3$ in the TTL show a large climatological uncertainty and SPARC/IO3C/GAW

(2019) recommends the use of in-situ observations. This is done here with NDACC ozonesondes. Against this dataset, the agreement is within ±10% between 50-200 hPa, increasing to 18% at 260 hPa. Note also the similar disagreement between BRAM2 and the independent observations from ACE-FTS, MIPAS and ozonesondes below 150 hPa. This suggests good agreement between these three observational datasets and a positive bias in MLS $O_3$ (around 30% at 260 hPa).

At 100 hPa, the standard deviation is around 25% against MLS, 35% against ozonesondes, 45% against MIPAS and 60% against ACE-FTS. Below 120 hPa, the standard deviation against MIPAS is the largest with values greater than 60%. The standard deviation against ACE-FTS is maximum around 147 hPa, around 70%, and decreases to 40% at 260 hPa. The agreement with ozonesondes is the best amongst the independent observations, 50% at 150 hPa and 40% at 200 hPa. Considering the limitations of BRAM2 and the satellite observations mentioned above, this level of agreement is
satisfactory.

Time series of the biases against MLS are stable even though they show seasonal variations (see Fig. 13), the highest amplitude being found at 100 hPa (∼15%). At 68 hPa, the ranges of the bias variations against the independent observations are small (<10%). Larger amplitudes are found at 100 and 147 hPa where the agreement against ozonesondes is usually within ±10 and ±15%, respectively. This is satisfactory considering the low spatial resolution of BASCOE and
the high vertical resolution of ozonesondes. The standard deviations of the differences against ozonesondes are stable over the years with significant variations (the peak to peak variations are around 10% at 68 hPa and 20% at 100 and 148 hPa). Considering the low abundance of $O_3$ in the TTL and the limitations of BASCOE and satellite measurements in this region, we found good agreement between BRAM2 and independent observations. Overall, we recommend BRAM2 $O_3$ in the TTL for scientific use.

**$H_2O$:** The bias against MLS is negligible above 100 hPa and is around −5% at 120 hPa (Fig. 12). At these levels, the bias is within the MLS accuracy. Below 120 hPa, the bias increases to 25% at 178 hPa. At altitudes above 200 hPa, similar biases are found against ACE-FTS and MIPAS: around 5% at 70 hPa, −10% at 100 hPa and around -30% at 200 hPa. Also, the biases against ACE-FTS and MIPAS are of opposite sign compared to the bias against MLS around 150 hPa,

which suggests that MLS has a negative bias against ACE-FTS and MIPAS. On the other hand, comparisons between MLS and in-situ Cryogenic Frost point Hygrometer (CFH) observations display similar bias profiles to those between MLS and BRAM2 (Vömel et al., 2007; Yan et al., 2016, respectively their Fig. 6 and 8). This suggests that BRAM2 is closer to in-situ CFH observations in the upper troposphere with MLS being too dry and ACE-FTS and MIPAS being too wet.

The standard deviation against MLS is around 15% at 100 hPa and increases in the upper troposphere, around 60% at 200 hPa. At altitudes below 100 hPa, the standard deviation of (BRAM2-MLS) exceeds the MLS precision. The standard deviations against ACE-FTS are significantly higher, around 25% at 100 hPa and 80% at 200 hPa. Standard deviations against MIPAS are higher than 100% at 200 hPa.

The biases against MLS are stable over time at 68 and 100 hPa (see Fig. 13). At 147 hPa, the time series of the bias

seems to increase (from ∼10% in 2005 to ∼15% in 2018) with small seasonal variations (amplitude around 5%). The bias against ACE-FTS shows higher variability, probably due to the low sampling of ACE-FTS in the Tropics. Against MIPAS, the biases are large at each level, with seasonal variations around 20%.

Overall, above the tropopause (approximately at 100 hPa in the Tropics), we recommend BRAM2 $H_2O$ for scientific use. In the upper troposphere BRAM2 overestimates MLS and underestimates ACE-FTS and MIPAS, and seems to be

in good agreement with CFH observations. Nevertheless, the standard deviations of the differences are large in the upper troposphere, and below 120 hPa BRAM2 $H_2O$ should not be used without consulting the BASCOE team.

**CO:**  Against MLS, the bias is negligible and is within the MLS accuracy (Fig. 12). Against ACE-FTS, the bias is around 25% at 70 hPa and decreases to −8% at 215 hPa. This bias is likely due to differences between MLS and ACE-FTS. Against MIPAS, the mean difference varies between 18% at 80 hPa and −30% at 215 hPa.

Standard deviations of the differences against MLS are at their minimum at 100 hPa (20%) and increase to 30% at 68 and 215 hPa. The standard deviations are smaller than the MLS precision at altitudes above 147 hPa and slightly greater at 215 hPa. The standard deviation against ACE-FTS is lower than against MLS, which suggests larger variability in MLS observations than in ACE-FTS. Against MIPAS, the standard deviation is usually greater than against MLS.

Time series of the bias against MLS are stable, showing almost no variations at 68, 100 and 147 hPa (Fig. 13). At these levels, the time series of the bias against ACE-FTS are noisy (no clear seasonal variations), with an amplitude around 15%. Against MIPAS, the time series are also noisy at 100 and 147 hPa with an amplitude around 10%. At 68 hPa, the time series display a clear seasonal variation with a 25% amplitude. Since the retrieval of MIPAS CO is done in log-space, their AKs are vmr-dependent such that their use in the comparison with BRAM2 would have reduced the apparent

discrepancies. The impact of MIPAS AKs for CO has not been tested and is left for future comparison. Nevertheless, BRAM2 CO is well characterized in the TTL and we recommend the product for scientific use.

**CH$_3$Cl:** Against MLS, the bias is negligible and within the MLS accuracy (Fig. 12). Against ACE-FTS, the bias varies between $-15\%$ and $-25\%$ between 70 and 147 hPa. The standard deviation of the differences against MLS is around 20%, in good agreement with the MLS precision. Against ACE-FTS, the standard deviation is around 25% at 70 hPa and

35% at 147 hPa, which is larger than the standard deviation against MLS.

The bias against MLS does not show significant variations over time at 68, 100 and 147 hPa (Fig. 13). At these levels, the time series of the bias against ACE-FTS are noisy with a small noticeable drift at 68 and 100 hPa (the origin of this drift, from MLS or ACE-FTS measurements, has not been identified). The amplitude of this "noise" is between 20 and 30%. Excluding trend studies, we recommend BRAM2 CH$_3$Cl in the TTL for scientific use.

## 5.5   Summary

The evaluation of BRAM2 is based on the comparison against assimilated MLS data and independent observations from ACE-FTS, MIPAS, MLS_N2O_640, SMILES ClO and NDACC ozonesondes. The evaluation has been done in four regions: the Middle Stratosphere (MS), the Lower Stratospheric Polar Vortex (LSPV), the Upper Stratospheric lower mesospheric Polar Vortex (USPV) and the Tropical Tropopause Layer (TTL).

In general, the mean differences between BRAM2 and MLS are negligible and within the MLS accuracy. The standard deviations of the differences are also generally within the MLS precision, except for O$_3$ in the MS. This means that in general, BRAM2 can be considered as a proxy for MLS. Each species is discussed individually below. The vertical range of validity of BRAM2 in the four evaluated regions is given in Table 1.





**Table 1.** Vertical ranges of validity (in hPa) of MLS v4 assimilated species and the corresponding BRAM2 products in the Middle Stratosphere (MS), the Lower Stratospheric Polar Vortex (LSPV), the Upper Stratospheric lower mesospheric Polar Vortex (USPV) and the Tropical Tropopause Layer (TTL). Abbreviations: Not Evaluated (N. E.), Not Assimilated (N. A.).

| Species | MLS v4 | BRAM2 | | | |
|---|---|---|---|---|---|
| | | MS | LSPV | USPV | TTL |
| $O_3$ | 0.02-261 | 4-100[1] | 10-100 | N. A. | 50-250 |
| $H_2O$ | 0.002-316 | 0.1-100 | 10-100 | 0.1-10 | 50-tropopause |
| $N_2O$ | 0.46-68 | 3-68 | 10-100 | 3-10 | N. A. |
| $HNO_3$ | 1.5-215 | 3-100 | 10-100 | 2-10 | N. E. |
| HCl | 0.32-100 | 0.46-100 | 10-100 | N. E. | N. E. |
| ClO | 1-147 | 1.5-20 | 10-100 | N. E. | N. E. |
| $CH_3Cl$ | 4.6-147 | 10-100[2] | N. E. | N. A. | 50-150 |
| CO | 0.0046-215 | N. E. | N. E. | 0.1-10 | 50-200 |

[1] $O_3$ has not been assimilated above 4 hPa (see text for details). Above that level, BRAM2 $O_3$ has not been evaluated and should not be used.
[2] In the Tropics. At mid-latitudes, the vertical range is 30-100 hPa

**$O_3$:** Ozone has been evaluated in the MS, LSPV and the TTL. The BASCOE CTM has an ozone deficit around 1 hPa ($-20\%$ vs. MLS, Skachko et al., 2016) which is also present in other models (SPARC, 2010, see their Fig. 6.17). For this reason, MLS $O_3$ has not been assimilated (and has not been evaluated) at altitude above (i.e. pressure lower than) 4 hPa. With the exception of the TTL, BRAM2 agrees well with MLS ($\pm5\%$), ACE-FTS ($\pm5\%$), MIPAS ($-8$ to $5\%$) and ozonesondes ($\pm10\%$), with substantial changes in the values depending on the region of interest. In the TTL, MLS profiles display unphysical oscillations that are smoothed in BRAM2, with an agreement against ozonesondes generally better than $\pm10\%$. In the upper troposphere BRAM2 underestimates MLS by around 10% at 260 hPa and overestimates ACE-FTS, MIPAS and ozonesondes by around 20% at 260 hPa. This suggests that MLS has a positive bias against the three other instruments.

**$H_2O$:** Water vapour has been evaluated in all four regions. Between the tropopause and the model lid (0.1 hPa), BRAM2 $H_2O$ agrees very well with MLS ($\pm2\%$) and ACE-FTS ($\pm10\%$). Except in the LSPV, the agreement with MIPAS is also good (better than $\pm10\%$). In the LSPV, a larger bias against MIPAS is found that could be reduced by using the MIPAS averaging kernels in the comparison. Below the tropical tropopause, BRAM2 has a positive bias against MLS (around 25%) and a negative bias against ACE-FTS and MIPAS (around -30% at 178 hPa). On the other hand MLS



underestimates in-situ Cryogenic Frost point Hygrometer (CFH) observations by around 25% which suggests a good agreement between BRAM2 and CFH as well as a positive bias of ACE-FTS and MIPAS against CFH.

**$N_2O$:** Nitrous oxide has been evaluated in the MS, LSPV and USPV. BRAM2 is well characterized by comparison against independent observations at altitudes below (i.e. pressures larger than) 3 hPa. In the MS, BRAM2 agrees well with MLS ($\pm 1\%$), ACE-FTS ($\pm 10\%$), MIPAS ($\pm 15\%$) and MLS_N2O_640 ($\pm 5\%$). Above that level, BRAM2 is poorly

characterized by comparison against observations where time series of mean differences are noisy. In conditions of low abundance of $N_2O$ encountered during the subsidence of the polar vortex (LSPV and USPV), BRAM2 overestimates independent observations but unnormalized differences are generally small. This study reveals a negative drift between BRAM2 and the three independent observational datasets which suggests a negative drift in the MLS $N_2O$ standard product. This issue is under investigation by the MLS team (Livesey and colleagues, in prep).

**$HNO_3$:** Nitric acid has been evaluated in the MS, LSPV and USPV. BRAM2 is well characterized by comparison against independent observations below 3 hPa. Below 10 hPa, BRAM2 agrees well against MLS ($\pm 2\%$), ACE-FTS and MIPAS ($-10$ to $1\%$ in both cases). From 10 to 3 hPa, the mean differences grow ($\pm 10\%$ against MLS) and exceed $\pm 50\%$ above 2 hPa for the three datasets. Above that level, MLS precision degrades (to around 0.6 ppbv i.e. 6 times the typical amount of $HNO_3$ at that level) and the constraint by the assimilated observations on BASCOE is weak. In the polar

vortex after denitrification, BRAM2 has a small negative bias against independent observations ($-5\%$). Despite the lack of ion chemistry and sources of mesospheric $NO_x$ in BASCOE, enhanced $HNO_3$ in the USPV is well represented in BRAM2.

**HCl:** Hydrogen chloride has been evaluated in the MS and LSPV. BRAM2 agrees well with ACE-FTS in MS between 0.4 and 100 hPa at mid-latitudes and between 0.4 and 50 hPa in the Tropics. At altitude above 5 hPa, MLS HCl is drifting (see

Sect. 5.1 for details) which results in a positive drift in the comparison between BRAM2 and ACE-FTS. In the LSPV when HCl is completely depleted by heterogeneous chemistry on PSCs, BRAM2 has a positive bias against independent observations which remains small when the unnormalized bias is considered ($<0.2$ ppbv).

**ClO:** Chlorine monoxide has been evaluated in the MS and the LSPV. In the MS, BRAM2 agrees well with independent observations even though it is poorly constrained by MLS observations. The constraint of MLS observations is stronger in LSPV under conditions of chlorine activation, when BRAM2 agrees well with independent observations.

**CH$_3$Cl:** Methyl chloride has been evaluated in the MS and the TTL. At altitudes below 10 hPa in the Tropics and 30 hPa in mid-latitudes, BRAM2 agrees very well with MLS ($\pm 5\%$). The agreement is good with ACE-FTS in the Tropics ($\pm 20\%$) and less good at mid-latitudes ($-60$ to $20\%$). Above these altitudes, BRAM2 has a positive bias against MLS, likely because the averaging kernels of MLS are not used. BRAM2 agrees well with MLS ($\pm 1\%$) and ACE-FTS ($-25$ to $-15\%$) in the TTL.

**CO:** Carbon monoxide has been evaluated in the USPV and the TTL. In the USPV, during descent of mesospheric CO, BRAM2 agrees well with MLS ($\pm 5\%$) and independent observations (typically 10% against ACE-FTS and MIPAS). In the TTL, the bias between BRAM2 and MLS is negligible ($\pm 2\%$). BRAM2 agrees reasonably well with ACE-FTS and MIPAS in the TTL: typical biases are, respectively, 25 and 18% around 70 hPa to $-8$ and $-30\%$ at 215 hPa.

According to the evaluation of BRAM2 with MLS and independent observations, we recommend scientific use of BRAM2 with the following limitations. The use of BRAM2 species should be restricted to their evaluated regions (see Table 1). In the MS, O$_3$, N$_2$O and HNO$_3$ should be used at altitude below, respectively, 4, 3 and 3 hPa. BRAM2 N$_2$O and HCl should be excluded from any trend studies. Methyl chloride should be used below 10 hPa in the Tropics and 30 hPa at mid-latitudes. In the Tropics at altitudes below 50 hPa, BRAM2 HCl can be used with the caveat of a positive bias with respect to independent observations.

## 6   Conclusions

This paper presents a new reanalysis of stratospheric composition produced by the Belgian Assimilation System of Chemical ObsErvations (BASCOE). It is based on the assimilation of measurements from the Microwave Limb Sounder (MLS), onboard the Aura satellite, of O$_3$, H$_2$O, N$_2$O, HNO$_3$, HCl, ClO, CH$_3$Cl and CO. BRAM2 (BASCOE Reanalysis of Aura MLS version 2) covers the period 2004-2017. The reanalysis is evaluated by comparison with independent observations from ACE-FTS,

MIPAS, MLS_N2O_640 (i.e. $N_2O$ retrieved from the MLS 640 GHz radiometer until 2013), SMILES ClO and NDACC ozonesondes. The evaluation of BRAM2 has been done in four regions: the Middle Stratosphere (MS), the Lower Stratospheric Polar Vortex (LSPV), the Upper Stratospheric lower mesospheric Polar Vortex (USPV) and the Tropical Tropopause Region (TTL). Only species which are relevant in the selected region have been evaluated. Moreover, while the BASCOE model includes 58 chemical species, only those constrained by the assimilated MLS species have been evaluated. Finally, the analysis

uncertainties based on the standard deviation of the ensemble state have not been evaluated in this paper. It will be the subject of a future study.

BRAM2 is well characterized by comparison against independent observations and is in most cases recommended for scientific use. One important limitation is reported here: the BASCOE model, as other models, suffers from an ozone deficit around 1 hPa where it underestimates MLS by $\sim$20%. Since the lifetime of $O_3$ at these altitudes is shorter than the revisit time of

MLS, approximately 12 hours between an ascending and a descending orbit, data assimilation cannot correct this bias. MLS $O_3$ profiles have thus not been assimilated (and have not been evaluated) at altitudes above (i.e. at pressures lower than) 4 hPa. Above that level BRAM2 $O_3$ should not be used.

The mean and standard deviation of the difference between BRAM2 and ACE-FTS have been compared to the differences between collocated profiles of MLS and ACE-FTS provided in published validation studies. The mean differences are in

general similar which means that most of the differences between BRAM2 and the independent observations are due to the differences between these datasets and MLS. The standard deviations of the difference (BRAM2-ACEFTS) are usually as good as or better than those from (MLS-ACEFTS). This suggests that the representativeness errors within BRAM2 are smaller than the sampling errors inherent in validation studies based on collocation of profiles.

A BASCOE control run (no assimilation, denoted CTRL) initialized by BRAM2 has been run for several months to assess

the added value of the assimilation compared to a pure model run. Spatial gradients across dynamical barriers are improved in BRAM2 with respect to CTRL. The representation of the LSPV in the presence of PSCs is also improved, in particular for $H_2O$, HCl and ClO. Subsidence in the polar vortex is improved thanks to the assimilation. The BASCOE system does not include mesospheric sources of CO, $H_2O$ or $NO_x$ nor ion chemistry to account for the formation of $HNO_3$ in polar winters. Nevertheless, the MLS observations provide a sufficient constraint to correct for these model biases. The BASCOE model also



lacks detailed tropospheric processes (chemistry, washout, convection) and, again, the MLS data provide a sufficient constraint

to correct for these biases.

BRAM2 also adds value to the observations. MLS $O_3$ profiles display unphysical oscillations in the TTL which are smoothed

in BRAM2, in good agreement with independent observations. It also allowed us to identify a positive drift in the MLS $N_2O$

standard product, retrieved from the 190 GHz radiometer in the v4 MLS retrieval, against measurements from ACE-FTS,

MIPAS or MLS_N2O_640. Since BRAM2 is usually not biased against MLS, this reanalysis could be used to study the biases

between MLS and other instruments and to derive a bias correction scheme for future versions of BRAM2. In the upper

troposphere, the comparison of BRAM2 with MLS and independent observations suggests that MLS $O_3$ is overestimated and

MLS $H_2O$ is underestimated.

This study also indicates several directions to improve the reanalysis for future versions. The first one is to increase the spatial

resolution and to improve several processes in the BASCOE CTM, in particular the convection and the PSC microphysical

scheme. Improving the photochemical scheme for $O_3$ could reduce the BASCOE ozone deficit. We believe that short-lived

species, like $O_3$ around 1 hPa or ClO in the middle stratosphere, should not be assimilated. For ozone above 4 hPa, a better

approach would be to use EnKF and MLS observations to optimize model parameters that control the abundance of ozone in

this region. Such a method still needs to be developed. Including realistic upper boundary conditions for CO, $H_2O$ and $NO_x$,

as well as implementing ion chemistry, would improve the system to represent the USPV region. Implementing the use of

the averaging kernels would improve the analysis for $CH_3Cl$ at mid-latitudes. It would also improve the comparison against

independent observations like MIPAS $H_2O$ in the TTL and in polar winter conditions. Bias correction to remove the vertical

oscillations in the MLS $O_3$ profiles, and to remove drift in $H_2O$, $N_2O$ and HCl would also improve the analysis.

BRAM2 is available to the scientific community and will be extended to later years observed by MLS in the near future.

*Data availability.* BRAM2 6-hourly gridded outputs are freely available to registered users on the BIRA-IASB ftp site. Only the MLS

assimilated species plus $Cl_2O_2$ are available. Access information is available at http://strato.aeronomie.be/index.php/2-uncategorised/6-

bram. BRAM2 ModAtObs files (see Sect. 3.4) are available upon request to the BASCOE team (quentin.errera@aeronomie.be).



*Author contributions.* QE led the development of the BASCOE data assimilation system, designed the configuration for BRAM2, realized its evaluation, produced the figures and wrote most of the text. SC, YC, JD and SS contributed to the development of BASCOE. MLS, KW, DH and MS contributed in the writing of the section dealing with the observations from, respectively, MLS, ACE-FTS, ozonesondes and SMILES. All authors reviewed the manuscript before submission.

*Acknowledgements.* The Atmospheric Chemistry Experiment, also known as SCISAT, is a Canadian-led mission mainly supported by the
5   CSA and NSERC. The ozonesonde data used in this publication were obtained from the Network for the Detection of Atmospheric Composition Change (NDACC) and are publicly available (see http://www.ndacc.org). We thank Bavo Langerok for preprocessing the ozonesonde data in the HDF-5 GEOMS format. Work at the Jet Propulsion Laboratory, California Institute of Technology, was done under contract with the National Aeronautics and Space Administration. Thanks to Farahnaz Khosrawi, Stefan Lossow, Michael Kiefer and Norbert Glatthor for helpful discussions on MIPAS data. Thanks also to Richard Ménard for helpful discussions about the EnKF method. QE also dedicates this
10  work to his friend William Lahoz who passed away in April 2019.





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
