# Peer review of "Technical Note: Reanalysis of Aura MLS Chemical Observations"

_Atmospheric Chemistry and Physics, 2019_

## Referee Comment (RC1) · Anonymous Referee #2 · 2 Jul 2019

**Reviewer #2 comments to 'Technical Note: Reanalysis of Aura MLS Chemical Observations' by Errera et al.**

This paper documents a new reanalysis data set of atmospheric chemical composition data (called BRAM2) covering the upper troposphere to the lower mesosphere produced by assimilating observation from the Microwave Limb Sounder (MLS) into the BASCOE system. The paper will make a useful reference paper for users of the BRAMH2 data set. It is well written and the authors thoroughly validate the BRAM2 reanalysis against a range of independent observations. I recommend the paper to be published as ACP Technical Note once my general comments and minor specific comments below have been addressed.

**General Comments**

Section 5 of the paper is very long. I suggest to shorten the text and to present some of the information in tables. This will make it easier for the reader, who will mainly use this report as a reference paper for the data set, to find the relevant information. In particular:

- Section 5.1, description on pages 22-27: Include a table that has the same layout as Figure 6 but contains mean bias +/- uncertainty calculated over the whole period in each box. This should then allow you to shorten the text.
- Section 5.2, description on pages 32-34: As above, include a table that has the same layout as Figure 8 but contains mean bias +/- uncertainty calculated over the whole period in each box. This should then allow you to shorten the text.
- Section 5.4, description on pages 41-44: As above, include a table that has the same layout as Figure 13 but contains mean bias +/- uncertainty calculated over the whole period in each box. This should then allow you to shorten the text.

Please add a comment in the paper on how the not assimilated species are affected by the assimilation. In particular, how does running several streams affect the not assimilated long-lived species? Do you see jumps in those fields? And if you do is this something that has an impact?

**Specific Comments**

p1, l12: Change '(SMILES), N2O' to '(SMILES) and N2O'

p3. l1: Change 'Atmospheric reanalysis is' to 'An atmospheric reanalysis is'

p3, l14: Change 'but also limited' to 'but again limited to'

p3, l15: Add a reference paper to 'The second generation of MLS'.

p3, l15: Change 'is operating since' to 'has been operating since'

p3, l18: change 'constituents are assimilated' toe 'constituents have been assimilated'

p3, l19: Add a reference paper for BASCOE here already.

p4, l7: change 'the data assimilation system' to 'the BASCOE data assimilation system'

p4, l20: change 'has been released' to 'was released' and add the year it was released in.

p5, l2: add a reference paper for MLS in the first sentence.

p5, l20: Why does the BASCOE CTM suffer from this o3 deficit? Please add a sentence that explains why.

p6, l22: Please spell out the acronym SMILES

p7, l1-2: change '(Waymark et al, 2013)' to 'Waymark et al. (2013)'

p7-8, Section 2.2.1: Please add some uncertainty ranges from the validation studies you refer to give some more concrete information about the quality of the ACE-FTS data, like you do for ozone sondes.

p8-9, Section 2.2.2: Please add some uncertainty ranges from the validation studies you refer to give some more concrete information about the quality of the MIPAS data, like you do for ozone sondes.

p9-10, Section 2.2.4: Please add some uncertainty ranges from the validation studies you refer to give some more concrete information about the quality of the SMILES data, like you do for ozone sondes.

p9, l6: change 'are quite similar between the two instruments' to 'were quite similar…'. Also, this statement is pretty vague. Can you give some numbers?

p9, l7: Change 'Those of CLO are..' to 'Those of CLO were..'

p10, l8: Change 'the CTM' to 'the BASCOE CTM'

p10, l11: change 'Huijnen et al.'s' to ' Huijnen et al. (2016)'

p10, l16: Change 'CTM results' to 'BASCOE CTM results'

p.11, l 10: see my general comment about the impact of the streams on the not assimilated species, especially long-lived'. Please add a sentence here mentioning how the not assimilated species link up.

p11, l10: change 'The three next' to 'The next three'

p12, l2; change 'BRAM2 have shown' to 'BRAM2 had shown'

p13, l20: You state 'The cause of this drift has not been identified…'. Are you sure it does not come from running several streams?

p18, l12: 'Note that BRAM2 will not be discussed in the extratrocial UTLS.' Please add a sentence to clarify why? Is it no good there, do you not have validation data,…?

p19, caption of Fig.4 : Change 'Daily zonal mean mean' to 'Daily zonal mean'

p21, Caption of Fig. 5. Please mention in the caption the data sets where the period isn't 2005-2017. Also, I noticed you are using American English spellings (e.g. color, gray). Should this be british English from an ACP report?

p22, l2: Is the variability in MLS larger than ACE because there are more MLS observations?

p23, Figure 6: CH3Cl at 46 hPa. Why is there the spike in 2010/11? Is this a data problem? Please comment on this when you discuss the Figure (p. 27 at the moment)

p25, l4: Change 'similar agreement is found' to 'Similar drifts are found'

p25, l16: You say 'These values are stable over time at 4.6 and 46 Pa', but there is a lot of variability, so I don't think you can call them stable. I would suggest to change the sentence to 'These values display significant seasonal oscillations….'. Also, please explain what is causing those seasonal differences.

p26, l5: Please add the year to the reference: Froidevaux et al. (2008)

p 26, l14: Please explain why there is the seasonal change in the bias.

p28, l3: Please change 'between 10 and 100 hPa' to 'between 100 and 10 hPa'

p29, Figure 7: Please add a row with CTRL-MLS and a row for STD(CTRL-ML) because it should show nicely the impact of the assimilation and the improvement of BRAM2 over CTRL.

p29, Caption Fig 7: Please change 'the mean differences between' to 'the mean relative differences in % between' and also mention in the caption that the standard deviation is in %.

p30,l10: Change 'to a reasonable extent' to 'to a reasonable magnitude'

p30, l12: Please explicitly spell out again what the 'above mentioned regions' are.

p32, l18: The sentence ' which is small given the relatively low amount of N2O' doesn't make sense as those error values are large relative to the low N2O values, perhaps change it to simply ' which is small'

p34, l13/14: Change 'On the other hand' to 'Nevertheless'

p34, l20: Change 'and between 60N-90N' to 'and averaged between 60N-90N)

p35, Caption Fig 9: Please change 'the mean differences between' to 'the mean relative differences in % between' and also mention in the caption that the standard deviation is in %.

p36, l3: Change 'the mean and standard deviation' to 'the mean relative difference and standard deviation'

p36, l14: Change '(see Sect. 5.1)' to '(see Figure 5, Sect 5.1)'

p36, l15: Please change 'The major difference is the smaller bias found' to 'The major difference is the smaller bias found in USPV'

p37, l1: Please change 'The standard deviations against MLS' to 'The standard deviations of CO against MLS'

p39, l14/15: The sentence 'To highlight the differences… are shown at 100 hPa.' doesn't make sense, unless you really mean that the reason for showing O3 on theta levels and the others on pressure levels is to high light the different?? I would sugget to remove the first part of the sentence and only say 'O3 is shown at 390K… while the other species are shown at 100 hPa.

p39, l17: Please add a sentence about the control here (e.g. you could move the sentence from p40, l2-3 here) and explain why the control has those large differences in the West Pacific. Is it because of the missing troposphere?

p39, l19: Change 'In Fig. 11' to 'In BRAM2 in Fig. 11'

p39, l21: Remove 'which is not the case for CTRL' as you already say 'In contrast to CTRL'

p39, l22; Please add at the end of the sentence 'around the eastern flank of the anticyclone.'

p40, l4: '…for the 2005-2017 period'. Please add that the MIPAS dataset ends in 2012.

p.44, Section 5.5: I think it is good to have a summary section, but at the moment it just seems to repeat a lot what is said in details in the earlier sections. However, if you include the tables I suggest above and shorten Section 5.1, 5.2, 5.4 it will make more sense to have such a summary section.

p45, l4: Change 'In the TTL, MLS profiles' to 'In the TTL, MLS O3 profiles'

p46, l1: Please add a reference paper for the CFH observations here.

p47, l21: Change 'the Aura satellite, of O3..' to 'the Aura satellite, namely O3…'

p47, l22: Add after 2004-2017 'and will be extended in the future.'

p48, l14: Please add the reference papers for the published validation studies.

p48, l19: Please add that the control run has been run for several months in 2009 and 2010

p19, l6: The sentence '.. to derive a bias correction scheme for future versions of BRAM2.' Would that be for the validation data sets or in case you wanted to assimilate datasets other than MLS? you say that BRAM2 is usually not biased against MLS so it can't be for MLS?

---

## Referee Comment (RC2) · Anonymous Referee #3 · 23 Aug 2019

This paper evaluates BRAM2 against both assimilated and independent observations for multiple species. Overall, I find the paper to be an impressive piece of work that is well written and well organized, and think the BRAM2 output will be an important dataset for analysis by researchers. I only have two minor issues that I bring up below, followed by a few technical points.

Main issues:

The paper is quite long, and the descriptions of comparisons where the authors break down each species is tedious because the authors essentially list a bunch of information that is already in the figures (e.g., satellite X agrees with BRAM2 within y%, at z hPa...). There is still useful descriptions in these sections of the overall comparison, but I would suggest minimizing the listing of mean differences/standard deviations

where the information is already in the figures

Why are comparisons done to the forecast and not the analysis? I think of the analysis step as the "best estimate" of the atmospheric state (and the main point of reanalyses in general), not the forecast. Could the authors at least explain this decision to do FmO a little better?

Technical points:

Page 7, lines 3-4: Can some reference be given for the v2.1 quality flags algorithm?

Page 7, lines 18-20: Rather than just saying the O3/WV products were validated, could the authors also state some high level findings from these studies (e.g., whether or not MIPAS had biases).

Page 8, line 14: Why not average the ozonesonde data to a vertical resolution more closely corresponding to the model? Later, on page 22 line 19 you mention that this difference in vertical resolution might cause larger standard deviations. I don't see the advantage of using ozonesonde data on such a high vertical resolution.

Page 9, lines 17-18: Can the authors state the approximate vertical resolution of the model? This is important so readers can understand how it compares to the assimilated observations.

Page 16, line 13: ACFTS -> ACE-FTS

Page 25, line 3: I'm not sure this result suggests BRAM2 could be used as a transfer function, although I don't dispute that BRAM2 could be useful as a transfer function. The N2O drift seems more like a "garbage in-garbage out" scenario where if you assimilate an obs that has an unphysical drift in it (i.e., MLS N2O) and thus all of the other measurements show an apparent drift relative to BRAM2.

Page 26, line 13-14: same comment as above

Conclusion section: BRAM2 just assimilates MLS, and the authors have done an admirable job in comparing to independent data. I this section they talk about potential improvements, but one improvement they don't discuss is if and how the assimilation would be improved by assimilating new sources of data (e.g., in the regions where MLS data are unreliable). Some discussion of this would be interesting if the authors are willing.

---

## Author Comment (AC1) · 4 Oct 2019

*We wish to thank the referees for their constructive review. Here below are our answers to their questions and comments, in italic green. After our replies, we have included the updated manuscript (and the updated supplement) and the manuscript tracking the changes between this new version and the ACPD version.*

**Reply to referee #2**

This paper documents a new reanalysis data set of atmospheric chemical composition data (called BRAM2) covering the upper troposphere to the lower mesosphere produced by assimilating observation from the Microwave Limb Sounder (MLS) into the BASCOE system. The paper will make a useful reference paper for users of the BRAM2 data set. It is well written and the authors thoroughly validate the BRAM2 reanalysis against a range of independent observations. I recommend the paper to be published as ACP Technical Note once my general comments and minor specific comments below have been addressed.

General Comments

Section 5 of the paper is very long. I suggest to shorten the text and to present some of the information in tables. This will make it easier for the reader, who will mainly use this report as a reference paper for the data set, to find the relevant information. In particular:

- Section 5.1, description on pages 22-27: Include a table that has the same layout as Figure 6 but contains mean bias +/- uncertainty calculated over the whole period in each box. This should then allow you to shorten the text.
- Section 5.2, description on pages 32-34: As above, include a table that has the same layout as Figure 8 but contains mean bias +/- uncertainty calculated over the whole period in each box. This should then allow you to shorten the text.
- Section 5.4, description on pages 41-44: As above, include a table that has the same layout as Figure 13 but contains mean bias +/- uncertainty calculated over the whole period in each box. This should then allow you to shorten the text.

*According to the referee's suggestion, these three tables have been created providing quantitative values of the FmO statistics while the text keeps only the most important information from Figs. 5, 6, 8, 10, 12 and 13.*

Please add a comment in the paper on how the not assimilated species are affected by the assimilation. In particular, how does running several streams affect the not assimilated long-lived species? Do you see jumps in those fields? And if you do is this something that has an impact?

*In the end of Sect. 5, an additional subsection "Note on the BRAM2 unobserved species" has been included with the following text:*

*"The above evaluation focuses on the eight species constrained by MLS observations while BRAM2 includes many others. Although it is beyond the scope of this paper, one can ask what is the value of BRAM2 unobserved species. For long-lived species (e.g. methane or the chlorofluorocarbons), small discontinuities appear in the troposphere at the stream transitions. This is due to the fact that each stream is initialized by a 20 year simulation with time dependent emissions of tropospheric source gases (see Sect. 3.2) while emissions are kept constant during the stream productions. For short lived species, the impact is mixed. For example, we found an improvement from CTRL to BRAM2 $NO_2$ when compared to MIPAS or*

*ACE-FTS in the lower stratosphere but a degradation in the upper stratosphere. Except for Cl$_2$O$_2$ which is closely related to ClO during the chlorine activation period, BRAM2 unobserved species are not delivered."*

Specific Comments

p1, l12: Change '(SMILES), N2O' to '(SMILES) and N2O'
*Done*

p3. l1: Change 'Atmospheric reanalysis is' to 'An atmospheric reanalysis is'
*Done*

p3, l14: Change 'but also limited' to 'but again limited to'
*Done*

p3, l15: Add a reference paper to 'The second generation of MLS'.
*Done*

p3, l15: Change 'is operating since' to 'has been operating since'
*Done*

p3, l18: change 'constituents are assimilated' toe 'constituents have been assimilated'
*Done*

p3, l19: Add a reference paper for BASCOE here already.
*Done*

p4, l7: change 'the data assimilation system' to 'the BASCOE data assimilation system'
*Done*

p4, l20: change 'has been released' to 'was released' and add the year it was released in.
*Done*

p5, l2: add a reference paper for MLS in the first sentence.
*Done*

p5, l20: Why does the BASCOE CTM suffer from this o3 deficit? Please add a sentence that explains why.
*The BASCOE model ozone deficit is already discussed in Sect. 3.1, §2.*

p6, l22: Please spell out the acronym SMILES
*Done*

p7, l1-2: change '(Waymark et al, 2013)' to 'Waymark et al. (2013)'
*Done*

p7-8, Section 2.2.1: Please add some uncertainty ranges from the validation studies you refer to give some more concrete information about the quality of the ACE-FTS data, like you do for ozone sondes.

*Done, see the updated Section 2.2.1.*

p8-9, Section 2.2.2: Please add some uncertainty ranges from the validation studies you refer to give some more concrete information about the quality of the MIPAS data, like you do for ozone sondes.

*Does the referee mean p7-8 instead of p8-9 (which is what we assumed)? Uncertainty ranges of MIPAS are now included in the manuscript. The updated text has been inserted in Sect 2.2.2.*

p9-10, Section 2.2.4: Please add some uncertainty ranges from the validation studies you refer to give some more concrete information about the quality of the SMILES data, like you do for ozone sondes.

*For ClO, comparison of SMILES observations with other instruments has been done only in the Antarctic vortex, at the end of the winter, which was already cited in P9L7-9 (Sugita et al., 2013). At mid-latitudes, a chemistry climate model simulation has been compared with SMILES and MLS (Akiyoshi et al., 2016, cited in P9L3-5), allowing to infer the agreement between SMILES and MLS. The whole paragraph of P9L3-9 has been rewritten as:*
*"Direct comparison of SMILES and MLS profiles has been done in the Antarctic vortex for the year 2009 with an agreement around ±0.05 ppbv, for ClO abundance less than 0.2 ppbv (Sugita et al., 2013). At mid-latitudes, a chemistry climate model simulation nudged towards a meteorological reanalysis was compared against SMILES and MLS (Akiyoshi et al., 2016). It shows an agreement within 10-20% with these two instruments in the middle and upper stratosphere, in good agreement with the values found between BRAM2 and SMILES (see Sect. 5.1)."*

p9, l6: change 'are quite similar between the two instruments' to 'were quite similar...'. Also, this statement is pretty vague. Can you give some numbers?

*See our response to the previous comment.*

p9, l7: Change 'Those of CLO are..' to 'Those of CLO were..' p10, l8: Change 'the CTM' to 'the BASCOE CTM'

*Done.*

p10, l11: change 'Huijnen et al.'s' to ' Huijnen et al. (2016)' p10, l16: Change 'CTM results' to 'BASCOE CTM results'

*Done.*

p.11, l 10: see my general comment about the impact of the streams on the not assimilated species, especially long-lived'. Please add a sentence here mentioning how the not assimilated species link up.

*See my reply to the second General Comment.*

p11, l10: change 'The three next' to 'The next three'

*Done.*

p12, l2; change 'BRAM2 have shown' to 'BRAM2 had shown'
*Done.*

p13, l20: You state 'The cause of this drift has not been identified...'. Are you sure it does not come from running several streams?
*We are almost sure that this drift is not caused by running several streams. If it would have been the case, we would have seen discontinuities in the time series of the error scaling factors (Fig. 2) at the transition time between the streams.*

p18, l12: 'Note that BRAM2 will not be discussed in the extratropical UTLS.' Please add a sentence to clarify why? Is it no good there, do you not have validation data,...?
*In fact, BRAM2 performs well in the ex-UTLS. Because the focus of the scientific community is largely on the TTL, and because the paper was already quite long, we had decided to skip it in the first submission. The ex-UTLS is now discussed in the revised manuscript while more attention is provided to the TTL and with only little additional text. FmO profiles, time series and tables corresponding to the ex-UTLS are provided in the supplement. Section 5.4 (TTL) has been renamed accordingly (UTLS).*

p19, caption of Fig.4 : Change 'Daily zonal mean mean' to 'Daily zonal mean'
*Done*

p21, Caption of Fig. 5. Please mention in the caption the data sets where the period isn't 2005-2017.
*Done*

Also, I noticed you are using American English spellings (e.g. color, gray). Should this be british English from an ACP report?
*This will be addressed by the language editor before publication in ACP.*

p22, l2: Is the variability in MLS larger than ACE because there are more MLS observations?
*We don't think so, for example the $N_2O$ standard deviations are similar for both instruments below 10 hPa while higher up, the standard deviations against MLS are larger (poorer) than against ACE-FTS while the number of available profiles does not change. See also for $H_2O$ or $HNO_3$.*

p23, Figure 6: CH3Cl at 46 hPa. Why is there the spike in 2010/11? Is this a data problem? Please comment on this when you discuss the Figure (p. 27 at the moment)
*Spikes in the $CH_3Cl$ time series at 46 hPa are due to (1) limited number of ACE-FTS profiles in October 2010 and February 2011 and (2) the fact that some of them display unphysical vertical oscillations. This is included in P28L10-12.*

p25, l4: Change 'similar agreement is found' to 'Similar drifts are found'
*"Similar agreement" refers to the whole section on $N_2O$ and not only to the previous sentence (this is a new §). Moreover, the previous sentence has been deleted, see the response to Ref#3 P25L3, so it is now clear what the sentence refers to.*

p25, l16: You say 'These values are stable over time at 4.6 and 46 Pa', but there is a lot of variability, so I don't think you can call them stable. I would suggest to change the sentence to 'These values display significant seasonal oscillations....'. Also, please explain what is causing those seasonal differences.

*We replaced the sentence by "These values are stable over the years …" The cause of the seasonal variations may be due to including observations belonging to the polar vortex or the tropical region, the 30°-60°N region not being a strict definition of the mid-latitudes of the northern hemisphere. This explanation has been included in the paper.*

p26, l5: Please add the year to the reference: Froidevaux et al. (2008)
*Done*

p 26, l14: Please explain why there is the seasonal change in the bias.
*This sentence has been removed according to the modifications done following your first general comment. Nevertheless, the reason of the seasonal variation is probably due to including observations belonging to the polar vortex or the tropical region, see also our reply to your comment on P25L16.*

p28, l3: Please change 'between 10 and 100 hPa' to 'between 100 and 10 hPa'
*Done*

p29, Figure 7: Please add a row with CTRL-MLS and a row for STD(CTRL-ML) because it should show nicely the impact of the assimilation and the improvement of BRAM2 over CTRL.
*These two rows have been added in the supplement (Fig. S5) since adding two rows in Fig. 7 will make it too large to fit the page size.*

p29, Caption Fig 7: Please change 'the mean differences between' to 'the mean relative differences in % between' and also mention in the caption that the standard deviation is in %.
*Done*

p30,l10: Change 'to a reasonable extent' to 'to a reasonable magnitude'
*Done*

p30, l12: Please explicitly spell out again what the 'above mentioned regions' are.
*Done*

p32, l18: The sentence ' which is small given the relatively low amount of N2O' doesn't make sense as those error values are large relative to the low N2O values, perhaps change it to simply ' which is small'
*Done*

p34, l13/14: Change 'On the other hand' to 'Nevertheless'
*Done*

p34, l20: Change 'and between 60N-90N' to 'and averaged between 60N-90N)

*Done*

p35, Caption Fig 9: Please change 'the mean differences between' to 'the mean relative differences in % between' and also mention in the caption that the standard deviation is in %.
*Done*

p36, l3: Change 'the mean and standard deviation' to 'the mean relative difference and standard deviation'
*Done*

p36, l14: Change '(see Sect. 5.1)' to '(see Figure 5, Sect 5.1)'
*Done*

p36, l15: Please change 'The major difference is the smaller bias found' to 'The major difference is the smaller bias found in USPV'
*Done*

p37, l1: Please change 'The standard deviations against MLS' to 'The standard deviations of CO against MLS'
*Done*

p39, l14/15: The sentence 'To highlight the differences... are shown at 100 hPa.' doesn't make sense, unless you really mean that the reason for showing O3 on theta levels and the others on pressure levels is to high light the different?? I would sugget to remove the first part of the sentence and only say 'O3 is shown at 390K... while the other species are shown at 100 hPa.
*The suggestion of the referee has been implemented.*

p39, l17: Please add a sentence about the control here (e.g. you could move the sentence from p40, l2-3 here) and explain why the control has those large differences in the West Pacific. Is it because of the missing troposphere?
*This § has been rewritten as:*
*"During boreal summer, the lower stratosphere is influenced by the anticyclonic circulation located above Asia which is associated with the Asian summer monsoon (e.g. Randel et al, 2013). In BRAM2 (see Fig.11), which agrees relatively well with MLS in this region, the anticyclone is marked by low O3 abundance and high abundances of H2O , CO and CH3Cl above Asia, indicating air of tropospheric origin. Also related to the Asian summer monsoon anticyclone is transport from mid-latitudes to the Tropics. This transport is marked by the O3 tongue (values ~400 ppbv) in BRAM2 starting in the northern east Pacific and ending above India, which denotes air of stratospheric origin around the eastern flank of the anticyclone (Randel et al., 2013, their Fig.4, although for a different year). Processes related to stratospheric chemistry are relatively well reproduced by CTRL, such as this ozone tongue. This is not the case for tropospheric processes where CTRL shows large disagreement against MLS in the Asian summer monsoon region or over the Western Pacific in the Tropics."*

p39, l19: Change 'In Fig. 11' to 'In BRAM2 in Fig. 11'

*Done*

p39, l21: Remove 'which is not the case for CTRL' as you already say 'In contrast to CTRL'
*Done*

p39, l22; Please add at the end of the sentence 'around the eastern flank of the anticyclone.'
*Done*

p40, l4: '...for the 2005-2017 period'. Please add that the MIPAS dataset ends in 2012.
*Done*

p.44, Section 5.5: I think it is good to have a summary section, but at the moment it just seems to repeat a lot what is said in details in the earlier sections. However, if you include the tables I suggest above and shorten Section 5.1, 5.2, 5.4 it will make more sense to have such a summary section.
*See my reply to the first General Comment.*

p45, l4: Change 'In the TTL, MLS profiles' to 'In the TTL, MLS O3 profiles'
*Done*

p46, l1: Please add a reference paper for the CFH observations here.
*Done*

p47, l21: Change 'the Aura satellite, of O3..' to 'the Aura satellite, namely O3...'
*Done*

p47, l22: Add after 2004-2017 'and will be extended in the future.'
*Done*

p48, l14: Please add the reference papers for the published validation studies.
*Done*

p48, l19: Please add that the control run has been run for several months in 2009 and 2010
*Done*

p49, l6: The sentence '.. to derive a bias correction scheme for future versions of BRAM2.' Would that be for the validation data sets or in case you wanted to assimilate datasets other than MLS? you say that BRAM2 is usually not biased against MLS so it can't be for MLS?
*The end of the sentence has been removed since this idea is also explained in P50L16-17.*

**Reply to referee#3.**

This paper evaluates BRAM2 against both assimilated and independent observations for multiple species. Overall, I find the paper to be an impressive piece of work that is well written and well organized, and think the BRAM2 output will be an important dataset for analysis by researchers. I only have two minor issues that I bring up below, followed by a few technical points.

Main issues:

The paper is quite long, and the descriptions of comparisons where the authors breakdown each species is tedious because the authors essentially list a bunch of information that is already in the figures (e.g., satellite X agrees with BRAM2 within y%, at z hPa...). There is still useful descriptions in these sections of the overall comparison, but I would suggest minimizing the listing of mean differences/standard deviations where the information is already in the figures.

*The descriptions of the comparisons have been rewritten where, following one of the suggestions of Referee#2, three tables have been created providing quantitative values of the FmO statistics and allowing us to only include the most significant information in the text.*

Why are comparisons done to the forecast and not the analysis? I think of the analysis step as the "best estimate" of the atmospheric state (and the main point of reanalyses in general), not the forecast. Could the authors at least explain this decision to do FmO a little better?

*Showing the differences between the analysis and assimilated observations would be misleading, it would only represent the (perfect) fit to the observations which depends on the observation error. Validating the forecasts against independent observations, i.e. observations that have not been assimilated yet, shows the quality of the BASCOE forecasts and the consistency between the background and observation error covariances.*

Technical points:

Page 7, lines 3-4: Can some reference be given for the v2.1 quality flags algorithm?

*Version 2.0 is described in Sheese et al. (2015, AMT, which is a new citation in the revised manuscript) and v2.1 is updated to deflag events which may have been erroneously identified as outlier, as in Sheese et al. (2017, already cited in the paper). The sentence is updated as follows:*
*"All ACE-FTS data used in this study were screened using the version 2.1 quality flags algorithm, which is modified from Sheese et al. (2015) in order to deflag events which may have been erroneously identified as outliers (e.g. Sheese et al. 2017)."*

Page 7, lines 18-20: Rather than just saying the O3/WV products were validated, could the authors also state some high level findings from these studies (e.g., whether or not MIPAS had biases).

*See our reply to Referee#2, p8-9, Sect. 2.2.2.*

Page 8, line 14: Why not average the ozonesonde data to a vertical resolution more closely corresponding to the model? Later, on page 22 line 19 you mention that this difference in vertical resolution might cause larger standard deviations. I don't see the advantage of using ozonesonde data on such a high vertical resolution.

*Some ozonesonde profiles have been degraded to the BASCOE vertical grid to check the impact of the ozonesonde vertical sampling on the FmO statistics. It turns out that the standard deviations are only slightly reduced so our interpretation of high standard deviation due to the high vertical resolution is no longer valid. Instead, we now believe that the higher standard deviation is due to the representativeness (i.e. vertical AND horizontal resolution) of the ozonesondes which is much higher than for the model and the satellite observations. So we changed P22L19 "…likely due to the higher vertical resolution of ozonesondes (100 m) against the model (1-3 km)." by "… likely due to the higher representativeness of ozonesondes compared to the model."*

Page 9, lines 17-18: Can the authors state the approximate vertical resolution of the model? This is important so readers can understand how it compares to the assimilated observations.

*The following information has been included:*
*"The vertical resolution is around 1 km at 100 hPa, 1.5 km in the middle stratosphere and increases to 5 km above 1 hPa. In the troposphere, the resolution is around 1.5 km."*

Page 16, line 13: ACFTS -> ACE-FTS
*Done*

Page 25, line 3: I'm not sure this result suggests BRAM2 could be used as a transfer function, although I don't dispute that BRAM2 could be useful as a transfer function. The N2O drift seems more like a "garbage in-garbage out" scenario where if you assimilate an obs that has an unphysical drift in it (i.e., MLS N2O) and thus all of the other measurements show an apparent drift relative to BRAM2.
*This sentence has been removed.*

Page 26, line 13-14: same comment as above.
*This sentence has been removed.*

Conclusion section: BRAM2 just assimilates MLS, and the authors have done an admirable job in comparing to independent data. In this section they talk about potential improvements, but one improvement they don't discuss is if and how the assimilation would be improved by assimilating new sources of data (e.g., in the regions where MLS data are unreliable). Some discussion of this would be interesting if the authors are willing.

*The following text has been added at the end of the conclusions:*
*"Additional observations could also be considered as long as they add value to MLS and they do not introduce spurious discontinuities. Ozone profiles from the Ozone Climate Change Initiative ($O_3$ CCI) could be considered once their biases are removed, which is not yet the case (Hubert et al., AMT, 2016). Total ozone column observations provided by $O_3$ CCI (Lerot et al., JGR, 2014) , 
[revised manuscript text omitted]

---

## Author Response (AR2)

Dear Jens-Uwe,

Thank you for your review. This new version of the manuscript includes the two technical corrections you suggested: (1) replacing "upper troposphere lower stratosphere" by "upper troposphere/lower stratosphere" (same change for the upper stratosphere/lower mesosphere) and (2) to add ranges in the scatter plots in Fig. 11. Note that the units for ozone in Fig. 11 are now in ppbv instead of pptv in the previous version.

Best regards,
Quentin.